# DeltaSM: Delta-Level Contrastive Learning with Mamba for Time-Series Representation

## Abstract

Self-supervised contrastive learning offers a compelling route to transferable time-series representations in label-scarce settings. Yet existing frameworks face a persistent trade-off between preserving fine-grained local dynamics at high temporal resolution and scaling to long sequences under practical compute constraints. Convolutional encoders often require deep stacks to retain rapid transitions, whereas Transformers incur quadratic cost in sequence length, making high-resolution long-context training expensive. Recent selective state-space models such as *Mamba* enable linear-time ($O(L)$) sequence modeling and offer a promising path to mitigate this bottleneck. However, their potential for *general-purpose* time-series representation learning remains underexplored; to our knowledge, prior Mamba-based contrastive learners have not been evaluated on the full UCR 2018 archive (128 datasets) under a unified protocol. We propose **DeltaSM** (*Delta-selective Mamba*), a self-supervised framework for univariate time series that reconciles efficiency and expressivity. DeltaSM integrates (i) a lightweight Mamba backbone, (ii) token-budget-constrained training, and (iii) a $\Delta$-level contrastive objective that counterbalances Mamba's smoothing tendency. Specifically, we apply curvature-adaptive weighting to first-order differences of the latent sequence, encouraging the encoder to emphasize informative local transitions without increasing computational cost. At inference time, we further augment the learned time-domain embeddings with explicitly extracted frequency-domain descriptors from the raw signal to improve expressivity at negligible overhead. Across all 128 UCR datasets, under **Protocol A**—a unified compute setting with a fixed number of optimization steps and a standardized downstream classifier—DeltaSM converges in seconds and achieves classification accuracy comparable to or better than strong baselines such as TS-TCC, TS2Vec, and TimesURL, using a single global configuration and a fixed pretraining-step budget (300 optimization updates per dataset). On a focused subset that includes long-sequence datasets under **Protocol B**—where baselines are allowed their recommended training budgets and hyperparameters while DeltaSM remains fixed as in Protocol A—DeltaSM reduces pretraining time by up to 184× while remaining competitive. Extensive ablations confirm that curvature-based weighting is crucial for suppressing noise while capturing local dynamics, and that inference-time frequency integration provides complementary gains with minimal additional cost.

## 1 Introduction

### 1.1 Background

Time-series data arise in a wide range of real-world systems, including physiological monitoring (ECG, EEG, wearable sensors), industrial and manufacturing processes, energy and environmental systems, and user interaction logs. Across these domains, improving downstream performance—e.g., classification, anomaly detection, segmentation, and forecasting—remains an important challenge. Although time-series measurements can often be collected at scale with little manual effort, obtaining high-quality labels is costly and typically requires domain expertise. Moreover, available labels are frequently task-specific and capture only a limited aspect of the underlying system. This gap between *abundant unlabeled data* and *scarce task-specific labels*

motivates *self-supervised representation learning*: we pretrain a general-purpose encoder from raw sequences and reuse the learned representations across tasks. Even when labeled benchmarks such as the UCR Time Series Archive (UCR) (Dau et al., 2019) are available, learning latent structure in a *label-agnostic* manner—capturing signal regularities that may not be fully exploited by single-task supervised training—is crucial for transferability and generalization to unseen tasks. In this study, we focus on univariate *classification* benchmarks in the UCR archive and use time-series classification accuracy as the primary metric to assess representation quality. At the same time, our pretraining objective and encoder design do not assume a particular classifier or label structure, and thus the learned embeddings are, in principle, applicable to other tasks such as anomaly detection and forecasting.

## 1.2 Related Work and Challenges

Contrastive learning (CL) (van den Oord et al., 2018; Chen et al., 2020) has emerged as a leading self-supervised paradigm for time-series data. CL constructs multiple views (positive pairs) from the same sample via data augmentation and trains an encoder to pull their representations together in the latent space while pushing apart representations of different samples (negative pairs). Efforts toward *universal* time-series representation learning have accelerated in recent years. Early approaches include T-Loss (Franceschi et al., 2019), which learns invariances via a dedicated objective, and TNC (Tonekaboni et al., 2021), which introduces Temporal Neighborhood Coding under stationarity assumptions. CoST (Woo et al., 2022) separately models trend and seasonal components. More general-purpose frameworks such as TS-TCC (Eldele et al., 2021), TS2Vec (Yue et al., 2022), and TimesURL (Liu & Chen, 2024) further advance contrastive time-series pretraining: TS-TCC combines temporal and contextual consistency, while TS2Vec supports arbitrary-length subsequences through hierarchical contrastive learning. In addition, TimesURL and InfoTS (Luo et al., 2023) aim to improve representation quality by preserving frequency characteristics or selecting augmentations using information-theoretic principles. Methods that explicitly connect time- and frequency-domain consistency, such as TF-C (Zhang et al., 2022), have also been actively studied.

Despite this progress, a key unresolved issue in general-purpose encoder design is the **trade-off between capturing fine-grained local dynamics and achieving computational scalability**. First, there is a **dilemma between high-resolution processing and computational efficiency**. In many time-series tasks, discriminative cues concentrate in local dynamics—rapid transitions or the onset/offset of events—rather than in the global shape of an entire signal. Capturing such cues requires preserving high temporal resolution without aggressive downsampling. However, many existing approaches rely on Transformers whose computational cost grows super-linearly with sequence length (e.g., $O(L^2)$), or on deep convolutional stacks that become expensive when high-resolution features must be maintained throughout the network. As a result, long-sequence training can become prohibitively slow, or resolution must be sacrificed. Second, there is a **lack of an explicit bias toward local changes**. Many contrastive objectives emphasize global consistency across entire sequences or subsequences, effectively treating time steps uniformly. Consequently, the degree to which the encoder focuses on local changes (e.g., first-order differences) is left to the data distribution and the inductive bias of the architecture, rather than being directly shaped by the objective. When global feature extraction is favored for efficiency, fine-grained transitions are at higher risk of being overlooked.

## 1.3 State-Space Models and Mamba for Time Series

State-space models (SSMs) are a sequence modeling paradigm that combines dynamical systems with neural network parameterization. In particular, S4 (Gu et al., 2022) and its variants can capture long-range dependencies via continuous-time state dynamics. Building on this line of work, Mamba (Gu & Dao, 2023) introduces a *selective* SSM with input-dependent parameters, achieving Transformer-level performance with linear-time ($O(L)$) complexity. While Mamba-based models have recently been explored for time-series tasks, they have largely focused on specific applications such as long-horizon forecasting. As far as we are aware, a comprehensive evaluation of a Mamba-based *general-purpose* encoder on the entire UCR archive under a standardized *self-supervised representation learning protocol* remains unexplored. In this work, we leverage Mamba's computational efficiency and its ability to model long-range dependencies by adopting it as a general-purpose backbone. At the same time, due to its integrator-like dynamics, SSMs can exhibit

a smoothing (low-pass) tendency, which poses an intrinsic challenge for capturing the steep local changes highlighted above.

### 1.4 Contributions

We propose DeltaSM (*Delta-selective Mamba*), a time-series representation learning framework that addresses the above trade-offs. DeltaSM combines a Mamba backbone with $\Delta$-level contrastive learning to **explicitly learn local dynamics while enabling high-resolution, long-sequence processing under realistic compute budgets**. Our main contributions are:

- **Complementary integration of Mamba and $\Delta$-level contrastive learning.** We first leverage Mamba's linear-time modeling to process high-resolution sequences efficiently, and then introduce a new objective that contrasts first-order temporal differences ($\Delta$-level features) of encoder outputs. This objective acts as a differentiator that counterbalances Mamba's smoothing tendency and explicitly emphasizes rapid transitions. Ablation studies show that an **adaptive weighting mechanism based on curvature (second-order differences)**, rather than naive differencing, is **crucial** for capturing informative transitions while suppressing noise.

- **Scalable training via token-budget constraints.** We introduce a simple *token-budget* scheme for the lightweight Mamba backbone, enabling compute-controlled and efficient training across datasets while limiting time and memory usage.

- **Fast convergence and strong accuracy under a fixed pretraining-step budget.** Across 128 datasets in the UCR archive, DeltaSM converges substantially faster than strong baselines such as TS-TCC, TS2Vec, and TimesURL under a fixed pretraining-step budget (300 optimization updates per dataset), while achieving comparable or better classification accuracy in seconds. We further show that integrating frequency information at inference time—outside the training loop—improves accuracy while keeping training costs minimal.

The remainder of the paper is organized as follows. Section 2 describes the DeltaSM framework, and Section 3 presents experimental settings and results, including ablation studies and qualitative embedding analyses. Section 4 provides a theoretical analysis of the empirical results, followed by conclusions in Section 5.

## 2 Proposed Method

This section presents DeltaSM, a self-supervised contrastive learning framework for *univariate* time series. DeltaSM combines (i) a lightweight Mamba encoder as a **linear-time backbone for high-resolution sequences**, (ii) a $\Delta$-level contrastive objective that counteracts the smoothing bias of state-space dynamics by emphasizing local changes, and (iii) **explicit frequency-feature integration at inference** to complement time-domain representations with negligible training overhead. Figure 1 summarizes the full pipeline, from multi-resolution view generation to contrastive pretraining and downstream featurization.

### 2.1 Problem Setting

Let $\mathcal{D} = \{(x^{(i)}, y^{(i)})\}_{i=1}^{N}$ be a labeled univariate time-series dataset, where $x^{(i)} \in \mathbb{R}^{L_i}$ is a variable-length sequence of length $L_i$, and $y^{(i)} \in \{1, \ldots, C\}$ is the class label. In self-supervised pretraining, labels are *not* used; we treat $\{x^{(i)}\}_{i=1}^{N}$ as unlabeled. We denote by $\mathcal{A}$ a stochastic augmentation distribution that produces two views of the same instance:

$$(x^{(i,1)}, m^{(i,1)}), \ (x^{(i,2)}, m^{(i,2)}) \sim \mathcal{A}(x^{(i)}).$$

Here, $m^{(i,v)} \in \{0,1\}^{L^{(i,v)}}$ is a binary mask indicating *valid* time steps after augmentation (1 for valid steps; 0 for padded *and* time-masked steps). For efficient batching, we right-pad each view to the maximum view

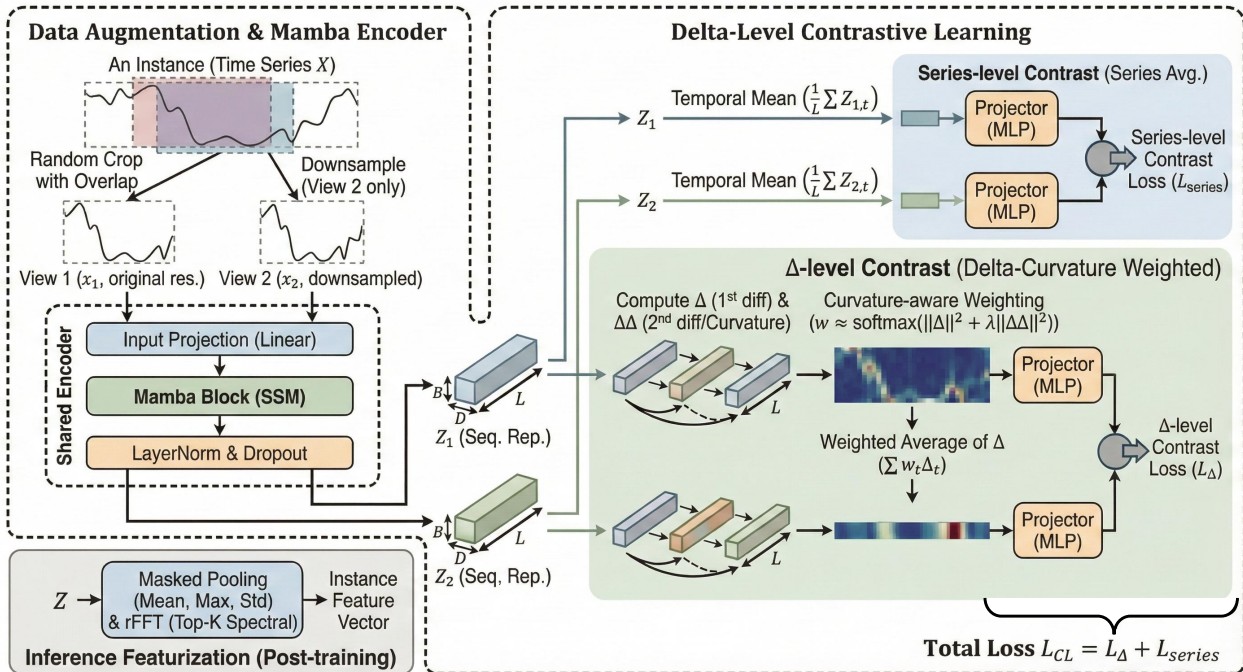

Figure 1: Overview of DeltaSM. We generate two correlated multi-resolution views, encode them with a shared Mamba backbone, optimize both $\Delta$-level and series-level symmetric contrastive losses, and construct downstream features by combining time-domain pooled representations with explicit frequency-domain descriptors at inference time.

length within the mini-batch and carry the corresponding masks. An encoder $f_\theta$ maps each view to a token-wise latent sequence:

$$z^{(i,v)} = f_\theta\left(x^{(i,v)}\right) \in \mathbb{R}^{L^{(i,v)} \times D}.$$

Contrastive learning then pulls together representations from the same underlying sequence and pushes apart representations from different sequences. In DeltaSM, we additionally apply contrastive learning to *first-order differences* of the latent sequence to explicitly promote sensitivity to local transitions.

## 2.2 Mamba Encoder and Token Budget

**Mamba encoder.** We adopt Mamba (a selective state-space model) as the backbone due to its linear-time complexity in sequence length. Given an input view $x^{(i,v)}$, we project each scalar observation into $D$ dimensions and process it with Mamba blocks:

$$u_t = W_{\text{in}} x_t \in \mathbb{R}^D, \tag{1}$$
$$h_{1:L} = \text{Mamba}_\theta(u_{1:L}), \tag{2}$$
$$z_t = \text{Dropout}(\text{LayerNorm}(h_t)). \tag{3}$$

Here, $t = 1, \dots, L$. In our default configuration, we use a lightweight setting and fix the embedding dimension to $D = 64$.

**Token-budgeted batch sizing.** Real-world time series exhibit substantial length imbalance across datasets and even within a dataset. To stabilize training under fixed physical resources (time/memory) and to equalize per-step throughput, we introduce a simple token-budget rule that selects the batch size adaptively from a global budget $B_{\text{tok}}$ and a batch-size cap $B_{\text{cap}}$. We also introduce two global length caps $L_{\text{cap}}^{\text{min}}$ and $L_{\text{cap}}^{\text{max}}$ that

bound the crop-length cap used in our view generation (see Sec. 2.3). In our default setting, we set the token budget to $B_{\text{tok}} = 32\,768$ and use length caps $L_{\text{cap}}^{\min} = 256$ and $L_{\text{cap}}^{\max} = 1024$. Let $\{L_i\}$ be the training-set lengths and let $L_{\max} = \max_i L_i$. We define a *reference length* $L_{\text{ref}}$ using a robust statistic:

$$L_{\text{ref}} \;=\; \max\!\Big(8,\; \min\big(L_{\text{base}},\; \text{percentile}_{90}(\{L_i\})\big)\Big).$$

Following the implementation, we set $L_{\text{base}} = L_{\text{cap}}^{\max}$ if $L_{\max} \geq \lfloor 1.5\, L_{\text{cap}}^{\max} \rfloor$, and $L_{\text{base}} = L_{\text{cap}}^{\min}$ otherwise (a conservative heuristic that avoids overestimating feasible batch sizes on datasets containing extremely long sequences). The batch size is then set as

$$B \;=\; \min\!\left(B_{\text{cap}},\, \max\!\left(8,\, \left\lfloor \frac{B_{\text{tok}}}{L_{\text{ref}}} \right\rfloor\right),\, N\right). \tag{4}$$

This enforces that the effective token throughput per step remains controlled (approximately $B \cdot L_{\text{ref}} \lesssim B_{\text{tok}}$), preventing pathological slowdowns on long-sequence datasets while keeping the training protocol stable and comparable across datasets.

**Temperature schedule.** For the temperature parameter $\tau$ in InfoNCE, we apply cosine annealing from an initial value $\tau_{\text{init}}$ to a final value $\tau_{\text{final}}$ over the pretraining steps to stabilize early optimization and encourage sharper discrimination later in training.

## 2.3 Multi-Resolution View Generation

Our augmentation $\mathcal{A}$ is designed to (i) preserve sufficient shared context between views for alignment and (ii) introduce resolution mismatch to encourage robust, transferable representations. Concretely, for each sampled instance of length $L_i$, we construct two correlated views via:

1. **Overlapping cropping (definition of $L_{\text{crop}}$).** We first determine a crop-length cap $L_{\text{cap}}$ based on the mini-batch statistics:

$$L_{\text{cap}} \;=\; \text{clip}\big(\text{percentile}_{80}(\{L_i\}_{\text{in batch}}),\; L_{\text{cap}}^{\min},\; L_{\text{cap}}^{\max}\big),$$

where $L_{\text{cap}}^{\min}$ and $L_{\text{cap}}^{\max}$ are global hyperparameters (introduced in Sec. 2.2). Then, for each instance we sample a crop ratio $r \sim \text{Uniform}(r_{\min}, r_{\max})$ (with $0 < r_{\min} \leq r_{\max} \leq 1$) and set the crop length as

$$L_{\text{crop}}^{(i)} \;=\; \min\big(\max(\lfloor rL_i \rfloor, 8),\; L_{\text{cap}}\big).$$

We extract two segments of length $L_{\text{crop}}^{(i)}$ from the original sequence, enforcing a minimum overlap ratio $\rho_{\text{ov}} \in (0,1)$ (implemented via constrained sampling of the second crop start index). In our default setting, we use $\rho_{\text{ov}} = 0.7$.

2. **Adaptive downsampling (definition of $L_{\text{target}}$).** To introduce multi-resolution views, we keep one crop at the original resolution and downsample the other. Let $L_{\text{target}}$ be the target token count (a hyperparameter; default 256). We choose a per-sample downsampling factor

$$d^{(i)} \;=\; \max\!\left(1,\, \left\lceil \frac{L_{\text{crop}}^{(i)}}{L_{\text{target}}} \right\rceil\right),$$

and apply simple average downsampling with stride $d^{(i)}$ to obtain a second view whose length is approximately $L_{\text{target}}$.

3. **Time masking and jittering.** To improve robustness to missingness and noise, we optionally mask a contiguous span in each view. With probability $p_{\text{mask}}$, we flip a consecutive block of length $\max(1, \lfloor \rho_{\text{mask}} L^{(i,v)} \rfloor)$ to invalid in the mask, where $L^{(i,v)}$ denotes the (augmented) view length and $\rho_{\text{mask}} \in (0,1)$ controls the span ratio. Masked positions are set to zero. We also add small i.i.d. Gaussian noise with standard deviation $\sigma_{\text{jit}}$ to *valid* time steps only.

## 2.4 Delta-Level Contrastive Learning

Mamba-based encoders inherit an integral (smoothing) inductive bias from state-space dynamics. To explicitly restore sensitivity to steep local transitions, DeltaSM introduces a $\Delta$-level contrastive objective that operates on *changes* in the latent sequence.

**Differential features and curvature-adaptive weights.** Given a latent sequence $z_{1:L}^{(i,v)} \in \mathbb{R}^{L \times D}$, we define the first- and second-order differences as

$$\Delta z_t = z_{t+1} - z_t \quad (t = 1, \ldots, L-1), \qquad \Delta^2 z_t = \Delta z_{t+1} - \Delta z_t \quad (t = 1, \ldots, L-2).$$

We assign a saliency score to each position (where curvature is defined) by

$$s_t^{(i,v)} = \|\Delta z_t^{(i,v)}\|_2^2 + \lambda_2 \|\Delta^2 z_t^{(i,v)}\|_2^2, \qquad t = 1, \ldots, L-2, \tag{5}$$

where $\lambda_2 \geq 0$ controls the curvature contribution. To avoid boundary artifacts induced by padding or time masking, we only score indices where the required tokens are valid. Concretely, we define the valid index set

$$\mathcal{T}^{(i,v)} = \left\{ t \in \{1, \ldots, L-2\} \mid m_t^{(i,v)} m_{t+1}^{(i,v)} m_{t+2}^{(i,v)} = 1 \right\}.$$

We then convert these scores into a probability distribution via a *mask-aware* softmax with *weight temperature* $w_{\text{temp}} > 0$:

$$w_t^{(i,v)} = \text{Softmax}_{t \in \mathcal{T}^{(i,v)}} \left( \frac{s_t^{(i,v)}}{w_{\text{temp}}} \right), \qquad w_t^{(i,v)} = 0 \text{ for } t \notin \mathcal{T}^{(i,v)}.$$

Finally, we aggregate a single $\Delta$-level summary vector by a weighted sum of first differences:

$$\tilde{\Delta}^{(i,v)} = \sum_{t \in \mathcal{T}^{(i,v)}} w_t^{(i,v)} \Delta z_t^{(i,v)} \in \mathbb{R}^D. \tag{6}$$

Intuitively, equation 5 de-emphasizes slowly varying regions and focuses the contrastive signal on salient transition points, while $w_{\text{temp}}$ controls how sharply the model concentrates on these points (large $w_{\text{temp}}$ approaches uniform weighting). Note that we score positions only for $t \leq L-2$ because the curvature term requires three consecutive tokens ($z_t, z_{t+1}, z_{t+2}$); in the unmasked case, uniform weighting therefore makes equation 6 telescope to an endpoint-difference statistic, up to boundary handling (see Section 4.1).

**Series-level summary.** In addition to $\tilde{\Delta}^{(i,v)}$, we also compute a series-level summary vector by mask-aware mean pooling:

$$\bar{z}^{(i,v)} = \frac{\sum_t m_t^{(i,v)} z_t^{(i,v)}}{\sum_t m_t^{(i,v)}} \in \mathbb{R}^D,$$

where $m_t^{(i,v)} \in \{0, 1\}$ denotes validity after padding and time masking.

**Loss function (symmetric InfoNCE).** Let $h$ denote either $\tilde{\Delta}$ or $\bar{z}$. We obtain projected, $\ell_2$-normalized representations by a projector $g_\phi$:

$$u^{(i,v)} = \frac{g_\phi(h^{(i,v)})}{\|g_\phi(h^{(i,v)})\|_2}.$$

To avoid view-asymmetry and to learn consistency bidirectionally across the two augmented views, we use the symmetric InfoNCE loss:

$$\mathcal{L}_{\text{Sym}}(u^{(1)}, u^{(2)}) = -\frac{1}{2B} \sum_{i=1}^{B} \left( \log \frac{\exp(s_{i,i})}{\sum_{k=1}^{B} \exp(s_{i,k})} + \log \frac{\exp(s_{i,i})}{\sum_{k=1}^{B} \exp(s_{k,i})} \right), \tag{7}$$

where $s_{i,j} = (u^{(i,1)})^\top u^{(j,2)}/\tau$ is cosine similarity scaled by temperature $\tau$ (Wang & Isola, 2020). We apply equation 7 to $\Delta$-level summaries and series-level summaries to obtain $\mathcal{L}_\Delta$ and $\mathcal{L}_{\text{series}}$, respectively. The final pretraining objective is the weighted sum

$$\mathcal{L}_{\text{CL}} = \mathcal{L}_\Delta + \alpha \mathcal{L}_{\text{series}}, \tag{8}$$

with balancing coefficient $\alpha$. We refer to $\mathcal{L}_{\text{series}}$ as the *series-level (instance-level)* contrastive loss since it is computed from a single pooled summary per sequence. To isolate the contribution of each component, we consider ablation variants: `No Delta` (remove $\mathcal{L}_\Delta$), `No Curv` (set $\lambda_2 = 0$), and `Uniform` (set $w_{\text{temp}}$ to a very large value, yielding near-uniform weights).

### 2.5 Feature Extraction and Frequency Integration for Downstream Tasks

**Explicit frequency integration.**   Frequency characteristics often provide complementary discriminative cues in time-series classification. However, enforcing spectral alignment inside the training loop can introduce nontrivial overhead. DeltaSM therefore **separates** time-domain contrastive pretraining from frequency-domain feature extraction: the encoder focuses on learning time-domain local dynamics, while frequency information is computed directly from raw inputs and appended *only at inference.*

**Feature vector construction.**   Given a frozen encoder $f_\theta$, we compute token-wise representations $z = f_\theta(x)$ and form time-domain statistics via masked pooling:

$$h_{\text{time}}(x) = \big[\text{mean}_m(z),\ \max_m(z),\ \text{std}_m(z)\big] \in \mathbb{R}^{3D}.$$

where the subscript $m$ indicates that padded/time-masked steps are excluded using the mask (and if no padding is present, $m \equiv 1$). We additionally compute frequency features $\phi(x) \in \mathbb{R}^K$ as the top-$K$ amplitudes from the rFFT magnitude spectrum of the *valid* (unpadded) raw sequence. If the spectrum contains fewer than $K$ bins (e.g., for very short sequences), we zero-pad $\phi(x)$ so that $\phi(x) \in \mathbb{R}^K$ always holds. The final fixed-dimensional feature vector is

$$\text{feat}(x) \ = \ \big[h_{\text{time}}(x),\ \phi(x)\big] \in \mathbb{R}^{3D+K}. \tag{9}$$

In our experiments, we set $K = 64$. This feature vector is then used by a downstream classifier (e.g., an SVM). Algorithm 1 summarizes the full training and inference pipeline.

## 3 Experiments

We evaluate the proposed framework, DeltaSM, by comparing it with representative self-supervised contrastive learning methods for time series. After describing the datasets, evaluation protocols, baselines, and implementation details, we report quantitative results, convergence behavior, component-wise ablations, and qualitative embedding visualizations.

### 3.1 Task and Datasets

We use the UCR Time Series Archive (2018) (Dau et al., 2019) for evaluation. The archive contains 128 univariate time-series classification datasets spanning diverse domains (e.g., physiological signals, sensor measurements, and image contours), with variable-length sequences and fixed train/test splits. We strictly follow the official splits and apply the following *unified preprocessing* to all datasets: (i) right padding to the maximum length within each dataset, (ii) conversion of missing values to NaNs and per-series $z$-normalization while ignoring NaNs, and (iii) zero imputation of any remaining NaNs. Padding positions are tracked by binary masks and are excluded from loss computation and feature extraction. Unless otherwise noted, we report test-set classification accuracy averaged over three random seeds.

### 3.2 Evaluation Protocols

To assess both *universality* and *efficiency* of DeltaSM, we use two complementary evaluation protocols.

**Protocol A: Universal Benchmark on 128 Datasets.**   We evaluate all 128 datasets using a *single global configuration* and a *fixed pretraining-step budget*. Each method performs self-supervised pretraining for a fixed number of optimization updates (300 steps) on the unlabeled training split. We then freeze the encoder and train an RBF-kernel SVM on the extracted representations, evaluating accuracy on the official test split.

---

**Algorithm 1** Training and Inference Pipeline of DeltaSM

---

**Require:** Unlabeled dataset $\mathcal{D} = \{x^{(i)}\}$, token budget $B_{\text{tok}}$, batch cap $B_{\text{cap}}$
**Require:** Encoder $f_\theta$, projector $g_\phi$, augmentation $\mathcal{A}$
 1: **Stage 1: Self-Supervised Pretraining**
 2: Compute reference length $L_{\text{ref}}$ (robust statistic over $\{L_i\}$)
 3: Set batch size $B \leftarrow \min\big(B_{\text{cap}}, \max(8, \lfloor B_{\text{tok}}/L_{\text{ref}} \rfloor), N\big)$
 4: **for** each training step **do**
 5:     Sample mini-batch $\{x^{(i)}\}_{i=1}^{B}$ from $\mathcal{D}$
 6:     $(x^{(i,1)}, m^{(i,1)}), (x^{(i,2)}, m^{(i,2)}) \sim \mathcal{A}(x^{(i)})$ {crop, downsample, mask, jitter}
 7:     $z^{(i,1)}, z^{(i,2)} \leftarrow f_\theta(x^{(i,1)}), f_\theta(x^{(i,2)})$
 8:     Compute saliency scores $s_t = \|\Delta z_t\|_2^2 + \lambda_2 \|\Delta^2 z_t\|_2^2$ and weights $w_t \propto \exp(s_t/w_{\text{temp}})$ over valid indices (mask-aware)
 9:     Aggregate $\tilde{\Delta}^{(i,v)} \leftarrow \sum_{t \in \mathcal{T}^{(i,v)}} w_t^{(i,v)} \Delta z_t^{(i,v)}$
10:     Compute series summaries $\bar{z}^{(i,v)}$ by masked mean pooling
11:     $\mathcal{L}_\Delta \leftarrow \mathcal{L}_{\text{Sym}}\big(\text{Normalize}(g_\phi(\tilde{\Delta}^{(1)})), \text{Normalize}(g_\phi(\tilde{\Delta}^{(2)}))\big)$
12:     $\mathcal{L}_{\text{series}} \leftarrow \mathcal{L}_{\text{Sym}}\big(\text{Normalize}(g_\phi(\bar{z}^{(1)})), \text{Normalize}(g_\phi(\bar{z}^{(2)}))\big)$
13:     Update $\theta, \phi$ to minimize $\mathcal{L}_\Delta + \alpha \mathcal{L}_{\text{series}}$
14: **end for**
15: **Stage 2: Inference & Feature Extraction**
16: **for** input sequence $x$ **do**
17:     $z \leftarrow f_\theta(x)$ {encoder is frozen}
18:     $h_{\text{time}} \leftarrow [\text{mean}_m(z), \max_m(z), \text{std}_m(z)]$ {mask-aware pooling}
19:     $h_{\text{freq}} \leftarrow \text{Top-}K(|\text{rFFT}(x)|)$ {from unpadded raw input; pad to $K$ if needed}
20:     **return** $\text{feat}(x) \leftarrow \text{Concat}(h_{\text{time}}, h_{\text{freq}})$
21: **end for**

---

To avoid ambiguity with the class-count notation $C$ (Section 2), we denote the SVM penalty parameter by $C_{\text{svm}}$ and fix $C_{\text{svm}} = 10$ with $\gamma = \texttt{scale}$. The compute budget is controlled by the fixed step count together with the token-budget-based batch-size constraint (Section 2.2). This protocol reflects resource-constrained deployments and measures how quickly each method acquires useful representations without dataset-specific hyperparameter tuning.

**Protocol B: Focused Comparison on Selected Datasets.** To complement Protocol A, we conduct a focused comparison on eight representative datasets: `Chinatown`, `ElectricDevices`, `PLAID`, `UWaveGestureLibraryAll`, `NonInvasiveFetalECGThorax1`, `NonInvasiveFetalECGThorax2`, `InsectWingbeatSound`, and `SemgHandMovementCh2`. These datasets cover diverse domains, scales, and sequence lengths (from 24 to about 1500 time steps), enabling a more detailed analysis. In Protocol B, baseline methods are allowed to use the training steps, batch sizes, and hyperparameters recommended in their original papers or public implementations. For TimesURL, we additionally allow downstream SVM hyperparameter selection (e.g., grid search), following its original evaluation protocol. In contrast, DeltaSM is evaluated under the same fixed setting as Protocol A (300 steps with fixed hyperparameters). This protocol therefore compares DeltaSM under a strict budget against baselines run in a near-recommended setting.

### 3.3 Baselines

We compare DeltaSM against three widely used contrastive learning baselines for time-series representation learning:

- **TS2Vec** (Yue et al., 2022): hierarchical contrastive learning for universal time-series representations.

- **TS-TCC** (Eldele et al., 2021): temporal and contextual contrasting for time-series pretraining.

- **TimesURL** (Liu & Chen, 2024): a frequency-aware framework with double Universum construction.

Table 1: Overall results on UCR-128 (Protocol A) with UCR official 1-NN baselines.

| Method | Mean acc. | Median acc. | Avg. Rank ↓ | Mean pretrain [s] | Median pretrain [s] | #Wins |
|---|---|---|---|---|---|---|
| 1-NN (ED) | 0.686 | 0.712 | 3.79 | – | – | 17 |
| 1-NN (DTW) | 0.728 | 0.754 | 3.03 | – | – | 32 |
| TS2Vec | 0.651 | 0.698 | 4.07 | 23.2 | 11.9 | 10 |
| TimesURL | 0.691 | 0.724 | 3.65 | 79.6 | 20.6 | 12 |
| TS-TCC | 0.671 | 0.686 | 3.84 | 14.0 | 7.7 | 17 |
| DeltaSM | **0.753** | **0.776** | **2.38** | **4.6** | **3.8** | **47** |

To isolate method effects as much as possible, we re-implement all baselines within a shared pipeline (data loading, preprocessing, and evaluation), switching only method-specific components.

### 3.4 Implementation Details

**Hardware.** All experiments are conducted on a workstation with an NVIDIA GeForce RTX 4090 (24 GB). We report wall-clock pretraining time (in seconds) measured in this environment.

**Pretraining Settings and Token Budget.** Unless stated otherwise, DeltaSM uses a single Mamba block and embedding dimension $D = 64$. To control per-update compute across datasets, we determine the batch size via the token-budget scheme in Section 2.2, using $B_{\text{tok}} = 32{,}768$ and $B_{\text{cap}} = 64$. In Protocol A, we also adjust batch sizes for baselines (within memory constraints) to keep the overall token processing comparable across methods.

**Optimization and Augmentation.** We use AdamW for all methods with learning rate $3 \times 10^{-4}$ and weight decay $10^{-4}$, together with a cosine learning-rate schedule. For DeltaSM, we use the multi-resolution two-view augmentation (cropping, adaptive downsampling, masking, and Gaussian noise) described in Section 2.3. For each baseline, we apply the augmentations recommended in the corresponding original work (e.g., random cropping and jittering).

### 3.5 Performance on the Entire UCR Archive (Protocol A)

Table 1 summarizes the results on all 128 UCR datasets under Protocol A. DeltaSM achieves a mean accuracy of 0.753 (75.3%) and a median accuracy of 0.776 (77.6%), yielding the best overall performance among the compared methods. It attains the best accuracy (including ties) on 47 datasets, compared with TimesURL (12), TS-TCC (17), and TS2Vec (10), indicating strong robustness across heterogeneous data distributions.

To compare methods across multiple datasets, we compute average ranks and assess statistical significance using a Friedman test followed by a Nemenyi post-hoc test at $\alpha = 0.05$ (Demšar, 2006). Figure 2 shows the resulting critical difference (CD) diagram.

DeltaSM is also substantially more efficient in wall-clock time. Its average pretraining time is 4.6 seconds, compared to 23.2 seconds for TS2Vec and 79.6 seconds for TimesURL. Figure 3 (Left) visualizes the accuracy–time trade-off under Protocol A, where DeltaSM consistently lies on the favorable frontier.

### 3.6 Focused Comparison on Selected Datasets (Protocol B)

Tables 2 and 3 report results for Protocol B: Table 2 summarizes overall performance, while Table 3 provides per-dataset accuracies. Under this setting—where baselines are allowed to follow recommended training budgets and hyperparameters—TS2Vec achieves the highest average accuracy (0.815; 81.5%). DeltaSM attains 0.789 (78.9%) on average, remaining competitive with TimesURL (0.803; 80.3%) and TS-TCC (0.744; 74.4%), despite using the fixed Protocol A configuration.

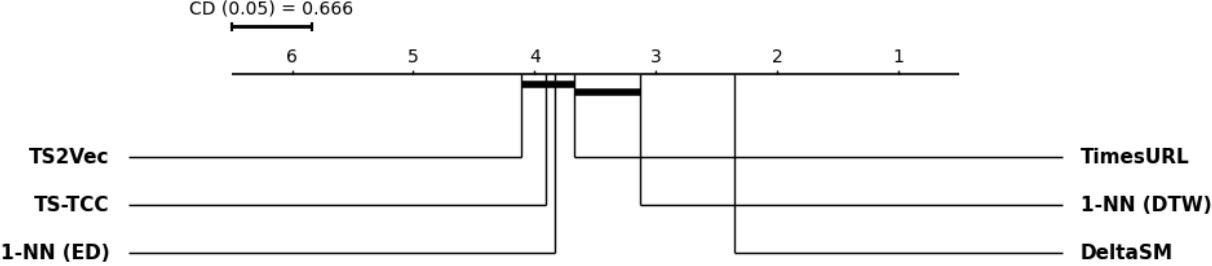

Figure 2: Critical Difference (CD) diagram on the 128 UCR datasets (Protocol A). The horizontal axis shows average rank (lower is better). Methods connected by a thick horizontal line are not significantly different.

Table 2: Overall results on 8 selected UCR datasets (Protocol B) with UCR official 1-NN baselines.

| Method | Mean acc. | Std. of acc. | Avg. Rank ↓ | Mean pretrain [s] | Std. of pretrain [s] |
|---|---|---|---|---|---|
| 1-NN (ED) | 0.702 | 0.226 | 4.50 | – | – |
| 1-NN (DTW) | 0.735 | 0.203 | 4.50 | – | – |
| TS2Vec | **0.815** | 0.165 | **2.19** | 18.32 | 9.69 |
| TimesURL | 0.803 | 0.172 | 2.31 | 771.65 | 367.75 |
| TS-TCC | 0.744 | 0.193 | 4.00 | 189.29 | 116.07 |
| DeltaSM | 0.789 | 0.147 | 3.50 | **4.20** | 0.73 |

Efficiency differences are more pronounced. DeltaSM averages 4.20 seconds of pretraining time, compared with 18.32 seconds for TS2Vec and 771.65 seconds for TimesURL (up to a 184× reduction). On long-sequence datasets such as `ElectricDevices` and `PLAID`, DeltaSM achieves both markedly shorter training time and strong accuracy (Figure 3, Right).

### 3.7 Learning Efficiency and Convergence Behavior

To ensure that DeltaSM's advantage under Protocol A is not merely an artifact of truncating training at 300 steps, we analyze test accuracy as a function of wall-clock training time. Figure 4 shows learning curves for DeltaSM and TS2Vec on `Mallat`, a representative long-sequence dataset.

We observe:

1. **Rapid convergence.** DeltaSM reaches a high-accuracy regime within the first second of training, indicating that the 300-step budget in Protocol A is sufficient for it to learn strong representations. In contrast, TS2Vec improves more slowly and remains in a substantially lower-accuracy regime even with longer training.

2. **Backbone–objective complementarity.** The fast convergence is consistent with a clear division of roles: the linear-time Mamba backbone efficiently captures broader context, while the $\Delta$-level loss provides an immediate inductive bias toward local transitions.

Overall, this analysis suggests that DeltaSM is particularly well-suited to compute-constrained settings where rapid representation acquisition is critical.

Table 3: Per-dataset accuracy on 8 selected UCR datasets (Protocol B) with UCR official 1-NN baselines.

| Dataset | 1-NN (ED) | 1-NN (DTW) | TS2Vec | TimesURL | TS-TCC | DeltaSM |
|---|---|---|---|---|---|---|
| ElectricDevices | 0.551 | 0.601 | 0.719 | 0.712 | 0.726 | **0.745** |
| InsectWingbeatSound | 0.562 | 0.355 | **0.623** | 0.611 | 0.394 | 0.481 |
| NonInvasiveFetalECGThorax1 | 0.829 | 0.790 | 0.923 | **0.945** | 0.879 | 0.894 |
| NonInvasiveFetalECGThorax2 | 0.880 | 0.865 | 0.935 | **0.946** | 0.900 | 0.916 |
| UWaveGestureLibraryAll | **0.948** | 0.892 | 0.932 | 0.937 | 0.736 | 0.824 |
| Chinatown | 0.953 | 0.956 | **0.974** | **0.974** | 0.945 | 0.918 |
| PLAID | 0.523 | 0.836 | 0.540 | 0.535 | **0.841** | 0.831 |
| SemgHandMovementCh2 | 0.369 | 0.584 | **0.873** | 0.765 | 0.530 | 0.705 |
| *Avg. Rank* | 4.50 | 4.50 | **2.19** | 2.31 | 4.00 | 3.50 |

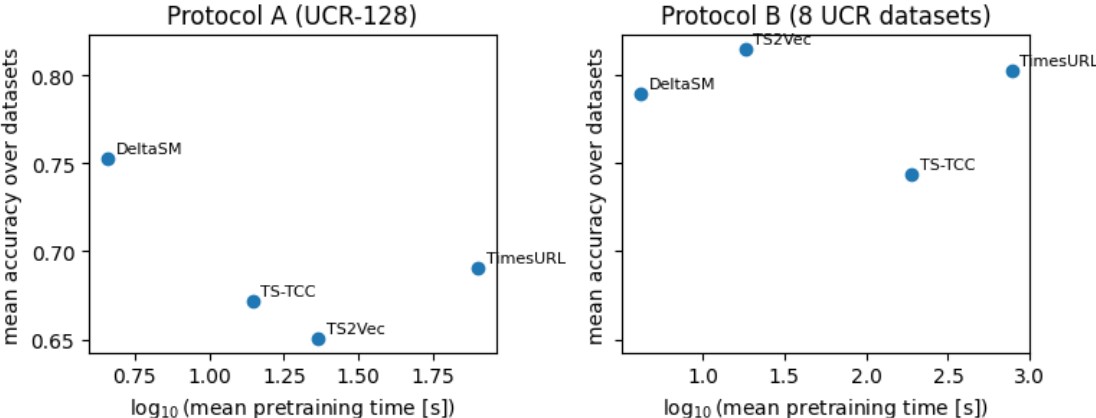

Figure 3: Relationship between average classification accuracy and average pretraining time. Left: Protocol A (UCR-128). Right: Protocol B (8 datasets). The horizontal axis shows training time on a logarithmic scale.

## 3.8 Ablation Study

We conduct ablations on the Protocol B datasets to quantify the contributions of DeltaSM components. We focus on (1) inference-time frequency feature integration and (2) training-time loss and backbone choices.

**Effect of Frequency Feature Integration.** Table 4 compares inference features under identical training settings (Time vs. Time-Frequency). Using only time-domain features yields an average accuracy of 0.768 (Base / Time), while concatenating frequency features improves performance to 0.792 (Base / Time-Frequency), i.e., a gain of more than two points. This supports the effectiveness of separating time-domain representation learning (training) from explicit frequency extraction (inference) to improve expressivity without increasing training cost.

**Loss Components.** Removing the first-order difference term (`No Delta`), removing curvature (`No Curv`), or enforcing uniform weights (`Uniform`) consistently degrades accuracy relative to the Base model. This indicates that, beyond naive differencing, curvature-adaptive weighting plays an important role in emphasizing informative transitions while suppressing noise.

**Backbone Encoders.** Table 5 compares backbones under the same loss and compute budget. Mamba achieves the highest average accuracy (0.785) and the shortest training time (4.43 s), compared to GRU (0.772, 4.63 s) and TCN (0.729, 5.14 s). These results highlight Mamba's linear-time sequence modeling as a key enabler of both speed and accuracy in this compute-controlled framework. While Mamba offers the best trade-off, the fact that all backbone variants are significantly faster than external baselines suggests that the primary driver of DeltaSM's efficiency is its lightweight framework design, including the token-budget scheme.

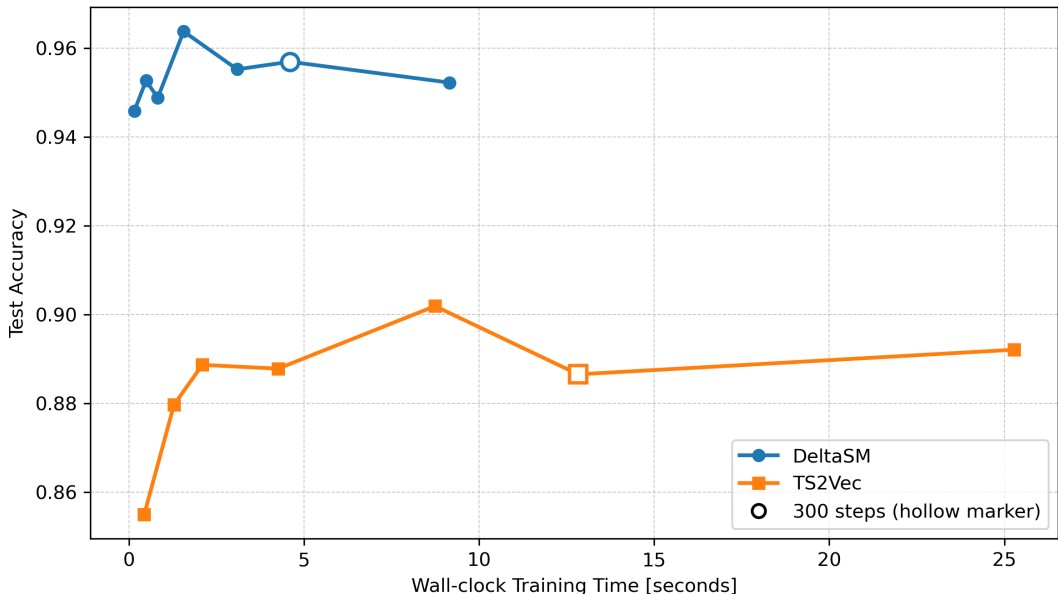

Figure 4: Convergence comparison on `Mallat`. DeltaSM exceeds 95% accuracy in roughly 0.5 seconds (about 30 steps) and remains stable thereafter, whereas TS2Vec improves more slowly and saturates around the high-80%–low-90% range even with longer training (hollow markers indicate 300 steps).

Table 4: Ablation study on loss components and inference feature domains.

| Avg. Acc. | Std. Acc. | Loss Type | Domain |
|-----------|-----------|-----------|--------|
| 0.768 | 0.192 | Base | Time |
| **0.792** | 0.166 | Base | Time-Frequency |
| 0.762 | 0.166 | No Curv | Time |
| 0.783 | 0.159 | No Curv | Time-Frequency |
| 0.750 | 0.194 | No Delta | Time |
| 0.779 | 0.162 | No Delta | Time-Frequency |
| 0.759 | 0.148 | Uniform | Time |
| 0.785 | 0.153 | Uniform | Time-Frequency |

### 3.9 UMAP Visualization of Embeddings

We qualitatively inspect learned representations using Uniform Manifold Approximation and Projection (UMAP) (McInnes et al., 2018). Figure 5 visualizes series-level embeddings on `TwoLeadECG` for models trained under Protocol A. While baselines show less structured separation under the short training budget, DeltaSM forms more compact and separated clusters, even before frequency feature integration. This suggests that DeltaSM can organize discriminative structure quickly in the representation space.

### 3.10 Summary

Across 128 UCR datasets under Protocol A, DeltaSM achieves strong average accuracy and consistently faster training than competitive baselines under a strict, unified compute budget. Under Protocol B, DeltaSM remains competitive with baselines run with recommended (and often larger) training budgets while retaining large efficiency gains. Convergence and ablation analyses further support that the combination

Table 5: Comparison of backbone architectures under identical loss and budget constraints.

| Backbone | Avg. Acc. | Std. Acc. | Avg. pretrain [s] | Std. pretrain [s] |
|---|---|---|---|---|
| Mamba | **0.785** | 0.165 | **4.43** | 0.92 |
| GRU | 0.772 | 0.166 | 4.63 | 1.56 |
| TCN | 0.729 | 0.139 | 5.14 | 1.70 |

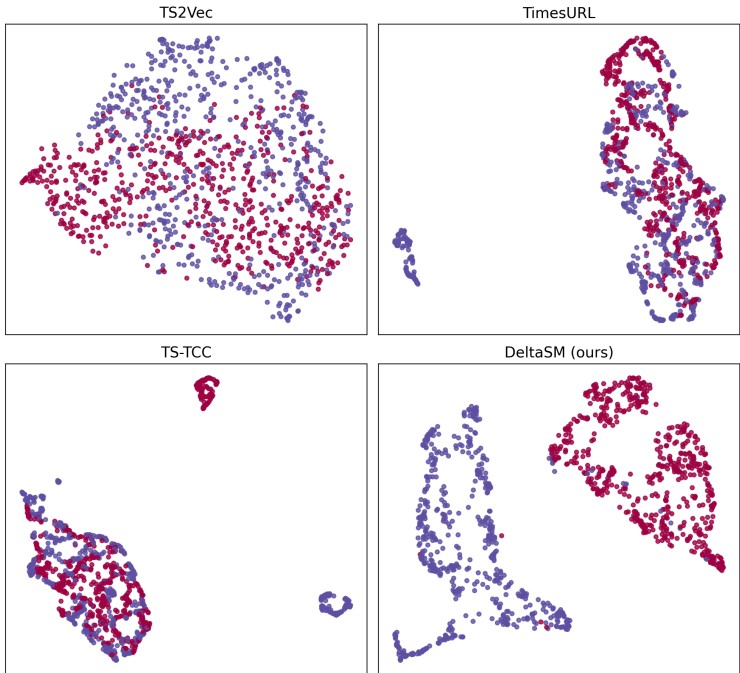

Figure 5: UMAP visualization of series-level embeddings on `TwoLeadECG` (Protocol A) (McInnes et al., 2018). DeltaSM forms clusters with less overlap between classes than competing methods under the same pretraining budget.

of a lightweight Mamba backbone, $\Delta$-level objectives, and inference-time frequency integration effectively mitigates the trade-off between high-resolution modeling and computational scalability.

## 4 Theoretical Analysis of Experimental Results

Section 3 showed that DeltaSM improves both classification accuracy and convergence speed under a fixed pretraining-step budget. This section links the empirical trends (ablation results in Table 4 and Table 5, and learning curves in Figure 4) to the three core design choices in Section 2: (i) curvature-adaptive reweighting of $\Delta$-level features, (ii) $\Delta$-level contrast paired with an SSM backbone (Mamba), and (iii) inference-time frequency augmentation.

### 4.1 Role of Curvature-Aware Weighting

A key component of DeltaSM is the $\Delta$-level summary $\tilde{\Delta}$ (equation 6), which aggregates latent first differences via a softmax distribution over saliency scores (equation 5). Table 4 shows that `Base` consistently outperforms both `Uniform` (near-uniform weights) and `No Curv` ($\lambda_2 = 0$), supporting the hypothesis that *not all local changes should contribute equally.*

**Why non-uniform weighting matters.** Let $z_{1:L}$ be the latent trajectory and $\Delta z_t := z_{t+1} - z_t$. Recall that in DeltaSM the weights are defined on the valid index set $\mathcal{T}$ (Section 2.4), which satisfies $t \leq L-2$ due to the curvature term. In the unmasked case $\mathcal{T} = \{1, \ldots, L-2\}$, if $w_t$ is uniform over $\mathcal{T}$, then the weighted sum collapses (up to a constant factor) to an endpoint-dominated statistic:

$$\tilde{\Delta} \; = \; \sum_{t \in \mathcal{T}} w_t \Delta z_t \; = \; \frac{1}{L-2} \sum_{t=1}^{L-2} (z_{t+1} - z_t) \; = \; \frac{z_{L-1} - z_1}{L-2},$$

(up to masking- and boundary-dependent conventions). Thus, `Uniform` discards information about *where* and *how* local transitions occur, and it can suffer from cancellation when $\Delta z_t$ oscillates in sign. In contrast, the softmax weighting in DeltaSM concentrates probability mass on a small subset of high-saliency positions, turning $\tilde{\Delta}$ into a transition-focused descriptor. This makes the $\Delta$-level contrastive loss informative even under our multi-resolution view generation (Section 2.3), where global alignment alone can be less sensitive to short-lived events.

**What curvature adds (and what it does not).** The curvature term $\|\Delta^2 z_t\|_2^2$ makes the saliency score sensitive to changes in the local slope of the latent trajectory. Compared with first-order energy alone, this biases the weighting toward regions that deviate from locally linear evolution (e.g., onsets/offsets and inflection points) and away from long, smoothly varying segments that can dominate the gradient budget by sheer volume. Crucially, this mechanism is still purely signal-based: it does not semantically distinguish meaningful events from spurious spikes. The empirical gain of `Base` over `No Curv` indicates that this simple shape-sensitive reallocation of learning signal is beneficial in practice under our unified training protocol.

## 4.2 Empirical Analysis of Integrator–Differentiator Complementarity

Section 2 frames DeltaSM as an "integrator–differentiator" pairing: an SSM backbone that efficiently aggregates long-range context and an auxiliary objective that explicitly emphasizes local changes. Both the backbone comparison (Table 5) and the loss ablations (Table 4) are consistent with this complementarity.

**Updated backbone evidence under Protocol B.** Revisiting the results in Table 5 (Section 3.8), under identical loss and budget constraints, Mamba achieves both the highest average accuracy and the shortest training time compared to GRU and TCN. This suggests that Mamba's efficient sequence modeling is particularly well-suited to serve as the "integrator" backbone that complements our differentiation-based objective.

**$\Delta$-level contrast counteracts low-pass and frequency bias in SSM backbones.** A (linear) continuous-time state-space model evolves as

$$\dot{\mathbf{s}}(t) = \mathbf{A}\mathbf{s}(t) + \mathbf{B}\mathbf{u}(t),$$

whose discretization induces an input–output mapping that can be implemented as either a recurrence or a convolution, with linear or near-linear scaling in sequence length (Gu & Dao, 2023). Recent analysis shows that a canonical structured SSM (continuous-time S4 under stable dynamics) is inherently low-pass and removes high-frequency components at each layer, making deep stacks prone to over-smoothing (Wang et al., 2025). Complementarily, SSMs can exhibit an implicit *frequency bias*, in which low-frequency content is learned more readily than high-frequency content; moreover, for popular SSM initializations, this bias is largely determined at initialization and is not corrected by standard training (Yu et al., 2025). These results motivate the hypothesis that, without explicit objective-level emphasis, an SSM encoder may underrepresent steep, short-duration transitions.

In DeltaSM, the $\Delta$-level objective provides exactly such emphasis. In particular, the first-difference operator is a discrete differentiator whose Fourier response scales as $|1 - e^{-i\omega}| = 2|\sin(\omega/2)|$, suppressing very low frequencies and emphasizing higher-frequency structure. Consistent with this view, removing $\mathcal{L}_\Delta$ (`No Delta`) substantially degrades performance relative to `Base` (Table 4), indicating that explicit supervision on latent differences is critical for preserving transition-sensitive information when using an SSM backbone.

**Why convergence is fast under a fixed budget.** The learning curves in Figure 4 suggest a division of labor across scales: the SSM backbone rapidly builds global context with linear-time computation (Gu & Dao, 2023), while the $\Delta$-level loss injects an immediate objective-level bias toward local dynamics via $\tilde{\Delta}$. Because $\mathcal{L}_\Delta$ is computed from the same forward pass as $\mathcal{L}_{\text{series}}$ (equation 8), this additional bias does not require a heavier backbone or longer training. In light of the documented tendency of SSMs to favor low-frequency structure (Yu et al., 2025), the explicit $\Delta$-level term provides a direct gradient signal for high-frequency/transition structure, offering a concrete explanation for the observed order-of-magnitude reduction in wall-clock time to reach a strong accuracy threshold (e.g., $\approx 90\%$ on Mallat) early in training.

### 4.3 Decoupling Time- and Frequency-Domain Information

Finally, Table 4 shows that augmenting time-domain embeddings with explicit frequency descriptors at inference (`Time-Frequency`) improves accuracy by roughly two points over the time-only setting (`Time`), without increasing contrastive pretraining cost. This supports a practical design principle under compute constraints: it is not necessary to force all discriminative cues into a single neural encoder during self-supervised training. Prior work on time-series self-supervision highlights the complementarity of time- and frequency-domain views and explicitly leverages them via time-frequency consistency or dual-domain objectives (Zhang et al., 2022; Woo et al., 2022). More generally, representation learning can benefit from incorporating non-neural, expert-designed descriptors alongside learned embeddings (Nonnenmacher et al., 2022). In DeltaSM, the encoder and $\Delta$-level objective are specialized for learning nonlinear temporal dynamics and transition structure, while global spectral information is obtained directly from the raw input via FFT and appended outside the training loop (Section 2.5). This decoupling increases representational capacity at negligible overhead, avoiding spectral modeling as a training bottleneck under the fixed-budget protocols in Section 3.

## 5 Conclusion and Future Work

We proposed **DeltaSM**, a universal representation learning framework designed to handle diverse time-series data under realistic compute constraints. DeltaSM combines (i) a linear-time Mamba encoder, (ii) a $\Delta$-level contrastive objective that explicitly targets local transition patterns, (iii) explicit integration of frequency-domain features at inference time, and (iv) token-budget-based adaptive batch control. Together, these design choices resolve the trade-off between "capturing detailed local information" and "computational scalability" in time-series representation learning, while ensuring fair compute usage across datasets.

**Key Findings and Contributions.** In a comprehensive evaluation on all 128 UCR datasets (Protocol A), DeltaSM achieved order-of-magnitude faster convergence than strong baselines such as TS2Vec and TimesURL under a fixed pretraining-step budget, reaching top-tier average accuracy within seconds. Additional analyses—including ablations under Protocol B and the discussion in Section 4—yield the following insights:

- **Significance of $\Delta$-level contrastive learning and curvature weighting:** Both the first-order differences of latent sequences and the **curvature-based adaptive weighting (second-order differences)** were confirmed to be key contributors to accuracy. This supports that the proposed objective goes beyond naive differencing by functioning as a filter that identifies and emphasizes the "quality" of local changes (e.g., steep transitions and inflection points), thereby counterbalancing Mamba's inherent smoothing tendency.

- **Superiority of the Mamba backbone:** On long-sequence datasets, Mamba exhibited a more favorable accuracy–efficiency trade-off than GRU and TCN. This suggests that linear-complexity sequence modeling with the capacity to preserve informative local changes over long contexts is highly suitable as a backbone for time-series contrastive learning. We also note that the primary driver of DeltaSM's overwhelming speed is its lightweight architecture anchored by the token-budget scheme.

- **Effectiveness of inference-time frequency integration:** Concatenating frequency-spectrum descriptors at inference time improves discriminability without increasing training cost. This validates

the **decoupling strategy** that assigns complementary roles to the time domain (training) and the frequency domain (inference).

**Limitations and Future Directions.** While this work takes an important step toward practical time-series foundation models, several directions remain open.

First, **extension to multivariate and irregularly sampled time series** is essential. The current DeltaSM assumes univariate sequences; incorporating channel interactions within Mamba blocks and introducing mechanisms (e.g., positional encodings) for irregular timestamps are important for broader real-world deployment (e.g., IoT sensors and medical logs).

Second, **broadening downstream tasks** is necessary. Although we focused on classification, evaluating how well the learned $\Delta$-aware representations transfer to anomaly detection, forecasting, and segmentation remains an important next step. In particular, it is theoretically expected that high sensitivity to local changes may be advantageous for anomaly detection.

Third, **rigorous convergence analysis** remains an open challenge. While we discussed the design from the perspective of inductive bias, deriving convergence guarantees and sample-complexity characterizations for the interaction between the $\Delta$-level objective and Mamba dynamics would provide deeper guidance for principled architecture design.

Finally, **advancing the adaptive weighting mechanism** is a promising direction. The current saliency weighting is heuristic; meta-learning approaches that learn which changes matter per dataset, or uncertainty-aware loss designs, could further improve robustness under noise and distribution shift.

Overall, DeltaSM provides a simple yet powerful baseline and a cornerstone toward scalable, accurate time-series representation learning under realistic compute budgets.

## Broader Impact Statement

This work primarily focuses on improving the efficiency and scalability of time-series representation learning. By significantly reducing the computational resources required for pretraining via the linear-time Mamba backbone and token-budget constraints, DeltaSM contributes to energy-efficient AI (Green AI) and lowers the barrier to entry for researchers and practitioners with limited hardware resources. However, as with any general-purpose representation learning framework, there is a potential for misuse in surveillance or privacy-invasive applications, particularly when applied to fine-grained personal data such as physiological signals or user activity logs. Furthermore, while DeltaSM demonstrates strong performance on standard benchmarks, its deployment in safety-critical domains (e.g., healthcare diagnostics or industrial control) warrants careful validation of robustness and reliability, as the current framework does not explicitly model uncertainty.

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

# A   Implementation Details and Theoretical Background for Symmetric InfoNCE

This appendix provides (i) a matrix-form implementation of the symmetric InfoNCE objective (equation 7) using standard cross-entropy primitives, and (ii) a brief information-theoretic perspective on InfoNCE and the role of the temperature parameter $\tau$.

## A.1   Matrix-Form Implementation via Cross-Entropy

Let $B$ denote the mini-batch size and let $d$ denote the dimensionality of the normalized representations used in contrastive learning (i.e., the output dimension of the projection head $g_\phi$). For each view $v \in \{1, 2\}$, we stack the $\ell_2$-normalized vectors $\{u^{(i,v)}\}_{i=1}^B$ row-wise:

$$U^{(v)} = \left[ (u^{(1,v)})^\top, \ldots, (u^{(B,v)})^\top \right]^\top \in \mathbb{R}^{B \times d}.$$

The pairwise similarity score matrix for all in-batch pairs is

$$S \;=\; \frac{1}{\tau} U^{(1)}(U^{(2)})^\top \in \mathbb{R}^{B \times B}, \tag{10}$$

whose $(i, j)$ entry equals $(u^{(i,1)})^\top u^{(j,2)}/\tau$. Let $y = (1, 2, \ldots, B)$ be the index-matching labels (the positive key for anchor $i$ is $j = i$). Then the two directional InfoNCE terms underlying equation 7 can be written as row-wise cross-entropy:

$$\mathcal{L}_{1 \to 2} = \mathrm{CE}(S, y), \qquad \mathcal{L}_{2 \to 1} = \mathrm{CE}(S^\top, y). \tag{11}$$

Accordingly, the symmetric objective is simply their average:

$$\mathcal{L}_{\mathrm{Sym}} \;=\; \frac{1}{2}\Big(\mathcal{L}_{1 \to 2} + \mathcal{L}_{2 \to 1}\Big) \;=\; \frac{1}{2}\Big(\mathrm{CE}(S, y) + \mathrm{CE}(S^\top, y)\Big). \tag{12}$$

This formulation is convenient in practice: (i) all pairwise scores are computed by a single matrix multiplication, (ii) numerical stability is handled by standard `logsumexp` implementations (e.g., PyTorch's `CrossEntropyLoss`), and (iii) it naturally supports dataset-dependent batch sizes B selected by the token-budget scheme (Section 2.2) without special-case code paths.

## A.2   Information-Theoretic Perspective and the Role of Temperature $\tau$

InfoNCE can be interpreted as a multi-class discrimination objective and is known to provide a lower bound on mutual information between two views under standard assumptions on negative sampling (e.g., negatives drawn i.i.d. from the marginal) (van den Oord et al., 2018; Poole et al., 2019). Let $\mathbf{u}^{(1)}$ and $\mathbf{u}^{(2)}$ denote the random representations of a positive pair. For batch size $B$ (with $B - 1$ in-batch negatives per anchor), a commonly used bound for the directional loss is

$$I\Big(\mathbf{u}^{(1)}; \mathbf{u}^{(2)}\Big) \;\geq\; \log B - \mathcal{L}_{1 \to 2}, \tag{13}$$

up to the assumptions and estimator details in the cited analyses. Since $\mathcal{L}_{\mathrm{Sym}}$ averages two directional InfoNCE terms, the mutual-information interpretation applies to each direction, while symmetrization typically yields more balanced gradients across the two views and improves optimization stability.

The temperature $\tau$ controls the sharpness of the softmax distribution over similarities. Smaller $\tau$ makes the distribution more peaked, increasing emphasis on hard negatives and promoting stronger alignment of positives, whereas overly small $\tau$ can cause unstable or saturated gradients. Following Section 2.2, we use cosine annealing to decay $\tau$ from $\tau_{\mathrm{init}}$ to $\tau_{\mathrm{final}}$, combining stable early training with a more discriminative objective later. Moreover, because the token-budget scheme induces dataset-dependent batch sizes B (fixed during pretraining for each dataset), tuning $\tau$ helps moderate loss-scale variation associated with different numbers of in-batch negatives.

## B    Experimental Details

### B.1    Hyperparameter Settings

Table 6 summarizes the training budgets and batch-size settings used in Protocol A (Universal Benchmark) and Protocol B (Focused Comparison). In **Protocol A**, the pretraining budget is defined as a fixed number of optimization updates—**300** steps for all methods. This budget is therefore *not* a wall-clock time limit (training time differs by method and is reported separately). In **Protocol B**, we allow baselines to follow the training steps, batch sizes, and hyperparameters recommended in the original papers or public implementations to approximate near-peak performance.

Table 6: Hyperparameter settings for DeltaSM and baselines across protocols.

| Method | Protocol A (Universal Benchmark) | | Protocol B (Focused Comparison) | |
|---|---|---|---|---|
| | Training Budget | Batch Size | Training Budget | Batch Size |
| **DeltaSM (Ours)** | Fixed **300** steps | Token budget ($B_{\text{tok}} = 32{,}768$), $B_{\text{cap}}=64$ | Fixed **300** steps | Token budget ($B_{\text{tok}} = 32{,}768$), $B_{\text{cap}}=64$ |
| TS2Vec | Fixed **300** steps | 16 | Auto (**200/600** steps) | 8 |
| TimesURL | Fixed **300** steps | 16 | Fixed **3000** steps | 8 |
| TS-TCC | Fixed **300** steps | 64 | Fixed **3000** steps | 64 |

*Evaluation / preprocessing:*
- Unified preprocessing as in Section 3.1. Protocol A uses a fixed RBF-SVM with $C_{\text{svm}} = 10$ and $\gamma = $ scale.
- In Protocol B, TimesURL is additionally allowed to tune downstream SVM hyperparameters (e.g., grid search), following its original evaluation.
- Optimizer / LR: AdamW with lr $3 \times 10^{-4}$, weight decay $10^{-4}$, and cosine learning-rate decay (Section 3.4).

## C    Full Results on the UCR Archive (Protocol A)

Table 7 reports classification accuracy (%) and self-supervised pretraining time (s) for all 128 datasets in the UCR Time Series Archive under Protocol A. Columns **1-NN(ED)** and **1-NN(DTW)** denote the accuracy of the 1-nearest-neighbor classifier with Euclidean distance and Dynamic Time Warping (DTW), respectively (DTW uses warping window $w = 100$). For each dataset, we bold the best accuracy across all methods and the fastest pretraining time among representation-learning methods.

Table 7: Full classification accuracy (%) and pretraining time (s) on 128 UCR datasets
(Protocol A). Best accuracy and best pretraining time (among representation learners)
are **bolded**.

| Dataset | 1-NN ED | 1-NN DTW | DeltaSM Acc. | DeltaSM Time | TS2Vec Acc. | TS2Vec Time | TimesURL Acc. | TimesURL Time | TS-TCC Acc. | TS-TCC Time |
|---|---|---|---|---|---|---|---|---|---|---|
| ACSF1 | 54.0 | 64.0 | 77.5 | **5.3** | **87.3** | 14.9 | 67.3 | 28.2 | 87.0 | 37.6 |
| Adiac | 61.1 | 60.4 | 70.2 | **3.3** | 74.6 | 17.2 | 54.8 | 19.1 | **74.8** | 7.1 |
| AllGestureWiimoteX | 51.6 | **71.6** | 64.3 | **3.4** | 63.4 | 9.8 | 46.7 | 26.6 | 55.9 | 9.7 |
| AllGestureWiimoteY | 56.9 | 72.9 | **73.5** | **3.4** | 72.7 | 8.7 | 57.4 | 25.3 | 63.4 | 9.8 |
| AllGestureWiimoteZ | 45.4 | **64.3** | 64.2 | **3.5** | 61.7 | 8.7 | 45.2 | 23.0 | 55.4 | 9.7 |
| ArrowHead | **80.0** | 70.3 | 75.6 | **3.1** | 79.6 | 10.5 | 74.5 | 16.5 | 66.3 | 6.7 |
| BME | 83.3 | 90.0 | 90.0 | 6.8 | **95.6** | 18.4 | 86.2 | 27.6 | 66.7 | **6.6** |
| Beef | **66.7** | 63.3 | 62.2 | **3.4** | 61.1 | 8.9 | 63.3 | 17.1 | 61.1 | 6.9 |
| BeetleFly | 75.0 | 70.0 | **89.9** | **3.3** | 71.7 | 9.9 | 60.0 | 17.1 | 75.0 | 6.4 |
| BirdChicken | 55.0 | 75.0 | 80.0 | **3.3** | **90.0** | 9.4 | 81.7 | 16.1 | 68.3 | 6.5 |
| CBF | 85.2 | **99.7** | 99.6 | 7.1 | 66.8 | 17.7 | 99.3 | 32.7 | 90.3 | **6.6** |
| Car | 73.3 | 73.3 | 70.0 | **4.7** | 72.2 | 9.9 | **78.3** | 17.6 | 69.4 | 11.7 |
| Chinatown | 95.3 | 95.6 | **95.8** | **6.0** | 93.3 | 14.0 | 94.0 | 26.4 | 95.8 | 12.2 |
| ChlorineConcentration | 65.0 | 64.8 | 62.1 | **3.3** | **73.4** | 16.5 | 60.9 | 23.7 | 64.2 | 7.2 |
| CinCECGTorso | 89.7 | 65.1 | 67.7 | **3.6** | 66.7 | 14.1 | **91.9** | 49.4 | 50.0 | 27.0 |
| Coffee | **100.0** | **100.0** | 92.9 | **3.2** | 89.3 | 13.0 | 89.3 | 18.1 | 89.3 | 6.7 |
| Computers | 57.6 | 70.0 | **72.2** | **5.1** | 63.6 | 12.0 | 57.5 | 19.6 | 67.5 | 14.7 |
| CricketX | 57.7 | **75.4** | 64.7 | **3.7** | 32.5 | 10.5 | 59.0 | 16.5 | 45.3 | 7.7 |
| CricketY | 56.7 | **74.4** | 67.2 | **3.9** | 29.4 | 9.5 | 61.2 | 14.4 | 49.3 | 7.7 |
| CricketZ | 58.7 | **75.4** | 68.4 | **3.7** | 33.5 | 9.5 | 59.2 | 14.2 | 39.0 | 7.7 |
| Crop | 71.2 | 66.5 | 66.6 | **5.4** | 56.4 | 13.6 | **73.2** | 28.2 | 70.9 | 7.8 |
| DiatomSizeReduction | 93.5 | **96.7** | 94.7 | **6.4** | 54.0 | 8.7 | 85.3 | 15.1 | 90.7 | 6.5 |
| DistalPhalanxOutlineAgeGroup | 62.6 | **77.0** | 72.4 | 7.5 | 71.9 | 16.8 | 70.0 | 30.5 | 70.5 | **7.3** |
| DistalPhalanxOutlineCorrect | 71.7 | 71.7 | 74.6 | 7.5 | 71.7 | 17.5 | **77.3** | 30.5 | 74.9 | **7.4** |
| DistalPhalanxTW | 63.3 | 59.0 | 64.3 | 7.5 | 62.4 | 17.5 | **65.7** | 33.0 | 65.2 | **7.4** |
| DodgerLoopDay | **55.0** | 50.0 | 49.2 | **3.7** | 31.2 | 10.2 | 53.3 | 15.9 | 47.9 | 7.8 |
| DodgerLoopGame | **88.4** | 87.7 | 87.0 | **6.1** | 52.2 | 9.9 | 86.5 | 16.5 | 59.9 | 6.8 |

Continued on next page

Table **7** – continued from previous page

| Dataset | 1-NN ED | 1-NN DTW | DeltaSM Acc. | DeltaSM Time | TS2Vec Acc. | TS2Vec Time | TimesURL Acc. | TimesURL Time | TS-TCC Acc. | TS-TCC Time |
|---|---|---|---|---|---|---|---|---|---|---|
| DodgerLoopWeekend | **98.6** | 94.9 | 96.9 | 7.5 | 85.7 | 9.3 | 95.7 | 15.5 | 92.3 | **6.7** |
| ECG200 | 88.0 | 77.0 | 82.3 | 3.7 | 64.7 | 16.4 | **92.7** | 32.6 | 84.7 | 7.3 |
| ECG5000 | 92.5 | 92.4 | **93.4** | 3.4 | 90.4 | 17.1 | 93.1 | 30.0 | 93.4 | 7.4 |
| ECGFiveDays | 79.7 | 76.8 | 75.7 | 6.5 | 54.6 | 17.3 | 85.1 | 32.0 | **95.0** | 6.8 |
| EOGHorizontalSignal | 41.7 | 50.3 | **51.8** | 5.2 | 38.5 | 15.0 | 45.4 | 25.1 | 47.0 | 29.6 |
| EOGVerticalSignal | 44.2 | **44.8** | 41.5 | 5.1 | 30.4 | 11.8 | 40.4 | 22.9 | 39.2 | 29.7 |
| Earthquakes | 71.2 | 71.9 | 63.1 | 4.1 | 74.6 | 11.6 | 74.8 | 17.2 | **75.5** | 10.3 |
| ElectricDevices | 55.1 | 60.1 | **75.6** | 3.5 | 64.6 | 15.6 | 63.8 | 32.6 | 68.0 | 7.3 |
| EthanolLevel | 27.4 | 27.6 | **38.0** | 3.9 | 27.3 | 16.3 | 28.7 | 101.2 | 29.6 | 47.8 |
| FaceAll | 71.4 | 80.8 | 67.8 | 3.6 | 42.2 | 16.5 | 75.0 | 13.1 | **81.3** | 7.2 |
| FaceFour | 78.4 | 83.0 | **84.4** | 3.6 | 28.0 | 9.4 | 57.2 | 15.8 | 48.1 | 6.5 |
| FacesUCR | 76.9 | **90.5** | 74.3 | 3.3 | 29.6 | 15.5 | 81.3 | 12.2 | 69.7 | 7.2 |
| FiftyWords | 63.1 | **69.0** | 57.6 | 3.7 | 33.0 | 11.0 | 62.3 | 13.6 | 43.5 | 7.5 |
| Fish | 78.3 | 82.3 | **86.0** | 4.0 | 73.9 | 9.0 | 72.0 | 16.1 | 66.7 | 9.6 |
| FordA | 66.5 | 55.5 | **81.1** | 4.1 | 76.4 | 9.7 | 73.4 | 16.3 | 77.4 | 9.7 |
| FordB | 60.6 | 62.0 | **68.2** | 4.1 | 61.4 | 8.6 | 64.5 | 15.9 | 65.4 | 10.0 |
| FreezerRegularTrain | 80.5 | 89.9 | 96.6 | 3.7 | 83.8 | 8.6 | 82.9 | 14.4 | **98.2** | 7.7 |
| FreezerSmallTrain | 67.6 | 75.9 | 75.8 | 3.2 | 75.4 | 7.6 | 72.2 | 13.5 | **87.9** | 6.7 |
| Fungi | 82.3 | 83.9 | **88.3** | 6.6 | 82.1 | 16.6 | 78.3 | 13.2 | 54.8 | **6.6** |
| GestureMidAirD1 | 57.7 | 56.9 | **67.9** | 3.5 | 66.2 | 8.2 | 61.8 | 12.0 | 62.8 | 8.3 |
| GestureMidAirD2 | 49.2 | **60.8** | 55.1 | 3.5 | 57.7 | 7.6 | 55.4 | 12.0 | 54.4 | 8.2 |
| GestureMidAirD3 | 34.6 | 32.3 | 35.4 | 3.5 | **42.6** | 7.5 | 41.8 | 11.9 | 30.0 | 8.3 |
| GesturePebbleZ1 | 73.3 | 79.1 | **82.5** | 3.7 | 72.3 | 8.1 | 79.5 | 12.6 | 46.9 | 9.4 |
| GesturePebbleZ2 | 67.1 | 67.1 | 71.5 | 3.8 | 71.5 | 8.2 | **76.2** | 12.5 | 44.3 | 9.4 |
| GunPoint | 91.3 | 90.7 | **97.3** | 3.3 | 95.3 | 18.6 | 91.6 | 12.7 | 95.8 | 7.0 |
| GunPointAgeSpan | 89.9 | 91.8 | **98.3** | 3.4 | 92.4 | 18.6 | 93.2 | 12.6 | 96.5 | 7.2 |
| GunPointMaleVersusFemale | 97.5 | **99.7** | 98.8 | 3.4 | 98.0 | 18.3 | 98.2 | 12.5 | 98.0 | 7.2 |
| GunPointOldVersusYoung | 95.2 | 83.8 | **96.3** | 3.4 | 95.9 | 18.0 | 95.4 | 12.6 | 92.8 | 7.2 |
| Ham | 60.0 | 46.7 | 64.4 | 4.0 | 59.7 | 7.9 | **72.4** | 15.2 | 52.1 | 8.9 |
| HandOutlines | 86.2 | 88.1 | **91.8** | 3.7 | 91.5 | 511.9 | 85.9 | 1243.4 | 69.4 | 95.7 |
| Haptics | 37.0 | 37.7 | **48.7** | 5.1 | 42.4 | 15.3 | 48.5 | 42.8 | 39.6 | 24.6 |
| Herring | 51.6 | 53.1 | **66.7** | 4.1 | 55.7 | 10.7 | 56.2 | 17.2 | 53.1 | 10.4 |
| HouseTwenty | 66.4 | 92.4 | **92.4** | 3.6 | 91.0 | 71.3 | 84.3 | 605.3 | 79.3 | 38.6 |
| InlineSkate | 34.2 | **38.4** | 37.5 | 3.7 | 35.8 | 40.5 | 24.7 | 387.4 | 32.1 | 54.9 |
| InsectEPGRegularTrain | 67.9 | 87.1 | **90.5** | 4.8 | 71.9 | 11.5 | 70.0 | 18.8 | 88.5 | 12.2 |
| InsectEPGSmallTrain | 66.3 | 73.5 | 74.3 | 3.2 | 73.5 | 9.2 | 71.9 | 16.3 | **79.3** | 6.6 |
| InsectWingbeatSound | 56.2 | 35.5 | 49.3 | 3.6 | 44.1 | 8.5 | **59.1** | 16.1 | 33.4 | 7.5 |
| ItalyPowerDemand | 95.5 | 95.0 | 85.6 | 6.7 | 91.6 | 14.5 | **96.1** | 29.6 | **96.1** | 9.7 |
| LargeKitchenAppliances | 49.3 | 79.5 | **86.3** | 5.0 | 71.0 | 10.4 | 50.6 | 18.4 | 79.5 | 14.7 |
| Lightning2 | 75.4 | **86.9** | 77.6 | 4.7 | 53.6 | 9.5 | 76.5 | 17.5 | 68.9 | 12.2 |
| Lightning7 | 57.5 | **72.6** | 70.3 | 3.7 | 28.8 | 9.0 | 61.2 | 15.0 | 64.8 | 7.8 |
| Mallat | 91.4 | **93.4** | 90.5 | 5.2 | 70.5 | 11.6 | 63.2 | 30.1 | 52.6 | 20.2 |
| Meat | 93.3 | 93.3 | **93.7** | 3.8 | 93.3 | 9.0 | 76.1 | 16.9 | 80.6 | 9.3 |
| MedicalImages | 68.4 | **73.7** | 66.1 | 3.3 | 53.0 | 17.9 | 65.2 | 33.5 | 69.8 | 7.2 |
| MelbournePedestrian | 84.8 | 79.1 | 79.8 | 7.5 | 75.2 | 14.6 | 85.3 | 27.6 | **88.5** | 11.1 |
| MiddlePhalanxOutlineAgeGroup | 52.0 | 50.0 | 51.3 | 7.5 | **58.9** | 17.4 | 53.4 | 31.7 | 49.8 | **7.4** |
| MiddlePhalanxOutlineCorrect | 76.6 | 69.8 | 73.3 | 7.5 | 74.6 | 17.1 | 72.5 | 33.0 | **77.7** | 7.4 |
| MiddlePhalanxTW | 51.3 | 50.6 | 51.9 | 7.5 | **57.1** | 16.8 | 54.8 | 32.6 | 50.2 | **7.4** |
| MixedShapesRegularTrain | **89.7** | 84.2 | 87.4 | 5.5 | 84.6 | 11.9 | 89.6 | 29.7 | 85.0 | 23.1 |
| MixedShapesSmallTrain | **83.5** | 78.0 | 82.0 | 5.4 | 75.6 | 11.0 | 81.5 | 28.8 | 76.1 | 23.1 |
| MoteStrain | **87.9** | 83.5 | 83.5 | 5.5 | 71.9 | 15.4 | 80.9 | 31.6 | 85.7 | 7.3 |
| NonInvasiveFetalECGThorax1 | 82.9 | 79.0 | **90.5** | 5.0 | 76.1 | 12.9 | 66.4 | 19.2 | 80.6 | 15.3 |
| NonInvasiveFetalECGThorax2 | 88.0 | 86.5 | **92.0** | 5.1 | 82.0 | 10.1 | 75.2 | 17.6 | 87.7 | 15.4 |
| OSULeaf | 52.1 | 59.1 | **62.8** | 4.0 | 44.5 | 9.7 | 43.5 | 15.7 | 56.7 | 9.0 |
| OliveOil | **86.7** | 83.3 | 84.4 | 3.4 | 81.1 | 9.0 | 44.7 | 16.6 | 53.3 | 7.7 |
| PLAID | 52.3 | 83.6 | **86.8** | 3.9 | 59.2 | 13.5 | 70.3 | 18.4 | 79.3 | 33.7 |
| PhalangesOutlinesCorrect | 76.1 | 72.8 | 78.8 | 5.8 | 72.3 | 15.3 | 70.9 | 31.3 | **80.7** | 7.2 |
| Phoneme | 10.9 | 22.8 | **26.0** | 5.4 | 15.7 | 12.4 | 18.7 | 31.2 | 15.3 | 23.1 |
| PickupGestureWiimoteZ | 56.0 | 66.0 | **67.5** | 3.3 | 48.0 | 9.4 | 61.3 | 26.9 | 59.3 | 7.5 |
| PigAirwayPressure | 5.8 | 10.6 | **19.7** | 3.7 | 14.1 | 71.7 | 7.7 | 605.5 | 2.9 | 60.3 |
| PigArtPressure | 12.5 | 24.5 | **74.2** | 3.6 | 64.6 | 69.4 | 9.5 | 630.1 | 5.1 | 60.2 |
| PigCVP | 8.2 | 15.4 | **36.1** | 3.9 | 11.2 | 70.1 | 4.3 | 638.4 | 3.2 | 60.4 |
| Plane | 96.2 | **100.0** | **100.0** | 3.7 | 84.1 | 16.8 | 97.1 | 24.7 | **100.0** | 7.2 |
| PowerCons | **93.3** | 87.8 | 87.4 | 3.3 | 79.4 | 18.0 | 90.2 | 20.8 | 81.9 | 7.2 |
| ProximalPhalanxOutlineAgeGroup | 78.5 | 80.5 | 78.9 | 6.1 | **86.3** | 16.9 | 85.2 | 30.8 | 83.9 | 7.2 |
| ProximalPhalanxOutlineCorrect | 80.8 | 78.3 | 85.3 | 7.6 | 80.4 | 17.4 | 79.7 | 32.2 | **85.9** | 7.3 |
| ProximalPhalanxTW | 70.7 | 75.6 | 76.7 | 7.5 | 78.0 | 17.5 | **79.3** | 33.3 | 78.0 | **7.3** |
| RefrigerationDevices | 39.5 | 46.4 | **53.4** | 5.0 | 48.0 | 12.1 | 43.9 | 20.4 | 50.3 | 14.6 |
| Rock | **84.0** | 60.0 | 59.3 | 3.3 | 64.7 | 614.1 | 78.0 | 2445.9 | 61.3 | 34.9 |
| ScreenType | 36.0 | 39.7 | **49.2** | 5.0 | 44.1 | 11.9 | 41.9 | 20.7 | 40.5 | 14.5 |
| SemgHandGenderCh2 | 76.2 | 80.2 | **85.7** | 3.6 | 62.8 | 14.2 | 78.3 | 318.5 | 76.2 | 38.8 |
| SemgHandMovementCh2 | 36.9 | 58.4 | **71.3** | 3.6 | 23.0 | 12.4 | 45.9 | 319.6 | 40.7 | 38.7 |
| SemgHandSubjectCh2 | 40.4 | 72.7 | **81.1** | 3.6 | 34.8 | 12.4 | 68.5 | 315.9 | 67.3 | 38.6 |
| ShakeGestureWiimoteZ | 60.0 | **86.0** | 82.0 | 3.5 | 74.0 | 11.9 | 78.0 | 29.8 | 74.0 | 7.8 |
| ShapeletSim | 53.9 | 65.0 | **78.4** | 3.3 | 52.6 | 9.3 | 52.8 | 17.6 | 52.0 | 6.4 |

Table **7** – continued from previous page

| Dataset | 1-NN ED | 1-NN DTW | DeltaSM Acc. | DeltaSM Time | TS2Vec Acc. | TS2Vec Time | TimesURL Acc. | TimesURL Time | TS-TCC Acc. | TS-TCC Time |
|---|---|---|---|---|---|---|---|---|---|---|
| ShapesAll | 75.2 | 76.8 | **78.7** | **4.1** | 75.2 | 9.4 | 61.0 | 16.1 | 57.3 | 10.2 |
| SmallKitchenAppliances | 34.1 | 64.3 | **78.0** | **4.9** | 76.4 | 10.5 | 62.9 | 18.5 | 66.8 | 14.6 |
| SmoothSubspace | 90.7 | 82.7 | 91.6 | **6.3** | 57.3 | 12.5 | 89.8 | 26.2 | **97.1** | 13.0 |
| SonyAIBORobotSurface1 | 69.5 | 72.5 | 75.2 | **7.2** | 42.9 | 16.6 | 73.2 | 33.0 | **77.1** | 13.9 |
| SonyAIBORobotSurface2 | **85.9** | 83.1 | 81.1 | **7.2** | 61.7 | 16.1 | 78.8 | 31.1 | 80.9 | 7.7 |
| StarLightCurves | 84.9 | 90.7 | **96.1** | **5.5** | 95.5 | 11.8 | 91.1 | 40.7 | 89.5 | 22.9 |
| Strawberry | 94.6 | 94.0 | 95.2 | **3.5** | **95.9** | 10.0 | 86.6 | 19.1 | 93.2 | 7.1 |
| SwedishLeaf | 78.9 | 79.2 | **87.4** | **3.3** | 66.4 | 17.8 | 82.7 | 34.9 | 84.1 | 7.1 |
| Symbols | 90.0 | **95.0** | 87.1 | **3.3** | 86.3 | 9.4 | 77.2 | 15.1 | 82.0 | 6.5 |
| SyntheticControl | 88.0 | **99.3** | 96.4 | **6.9** | 62.8 | 15.3 | 97.2 | 28.9 | 94.3 | 7.4 |
| ToeSegmentation1 | 68.0 | 77.2 | **83.3** | **3.7** | 70.6 | 8.6 | 63.6 | 14.5 | 63.5 | 6.7 |
| ToeSegmentation2 | 80.8 | **83.9** | 78.5 | **3.3** | 69.0 | 7.5 | 79.0 | 13.9 | 65.4 | 7.0 |
| Trace | 76.0 | **100.0** | 99.0 | **3.6** | 97.7 | 8.0 | 72.3 | 14.8 | 83.7 | 7.5 |
| TwoLeadECG | 74.7 | 90.4 | **95.9** | **6.2** | 64.2 | 15.9 | 72.4 | 32.4 | 77.0 | 7.2 |
| TwoPatterns | 90.7 | **100.0** | 82.3 | **3.3** | 36.6 | 17.2 | 74.4 | 33.6 | 69.8 | 7.1 |
| UMD | 76.4 | **99.3** | 95.6 | 7.4 | 75.9 | 17.4 | 76.9 | 28.9 | 93.3 | **6.9** |
| UWaveGestureLibraryAll | **94.8** | 89.2 | 90.0 | **5.3** | 78.9 | 11.4 | 87.9 | 27.4 | 53.8 | 20.0 |
| UWaveGestureLibraryX | **73.9** | 72.8 | 66.3 | **3.7** | 70.7 | 11.3 | 73.2 | 15.4 | 54.8 | 7.7 |
| UWaveGestureLibraryY | **66.2** | 63.4 | 59.2 | **3.8** | 64.6 | 7.7 | 65.1 | 13.1 | 49.7 | 7.7 |
| UWaveGestureLibraryZ | 65.0 | 65.8 | 65.1 | **3.7** | 66.0 | 7.9 | **67.5** | 13.0 | 52.0 | 7.7 |
| Wafer | **99.6** | 98.0 | 98.6 | **3.4** | 97.4 | 17.7 | 98.8 | 23.1 | 99.5 | 7.2 |
| Wine | 61.1 | 57.4 | 66.7 | **3.5** | **74.7** | 10.6 | 57.4 | 14.9 | 59.3 | 7.1 |
| WordSynonyms | 61.8 | **64.9** | 41.4 | **3.7** | 36.1 | 8.2 | 49.9 | 14.2 | 37.4 | 7.5 |
| Worms | 45.5 | 58.4 | **67.9** | **5.2** | 60.2 | 11.2 | 57.1 | 26.3 | 54.5 | 19.4 |
| WormsTwoClass | 61.0 | 62.3 | **75.8** | **5.2** | 71.0 | 10.7 | 59.7 | 23.0 | 63.6 | 19.4 |
| Yoga | 83.0 | **83.6** | 83.5 | **3.9** | 77.2 | 9.1 | 74.0 | 15.8 | 73.2 | 8.9 |

