# OpenReview forum: "DeltaSM: Delta-Level Contrastive Learning with Mamba for Time-Series Representation"
_TMLR — Withdrawn by Authors_

### Review · Reviewer_jHNS · 2026-02-09

**Summary Of Contributions:**

In this paper, the authors study self-supervised learning for time-series data with a Mamba-based approach. To this end, they explore different methods for introducing different views of time-series data, which they feed through a Mamba encoder to obtain processed representations. These representations are aligned using the infamous InfoNCE loss.

Strengths:

+ Self-supervised learning of representation for time-series data is a fundamental problem.


Weaknesses:

1. The paper has limited novelty and technwical contribution. The paper admits that the literature has alreayd explored a Mamba-based self-supervised learning approach for time-series data and argue that "However, their potential for general-purpose time-series representation learning remains underexplored; to our knowledge, prior Mamba-based contrastive learners have not been evaluated on the full UCR 2018 archive (128 datasets) under a unified protocol." Unfortunately, this does not suffice as a novel contribution.

2. Many claims, design choices or arguments are not justified:

2.1. "a self-supervised framework for univariate time series that reconciles efficiency and expressivity." => The paper makes a claim of expressivity in Introduction. However, it does not evaluate this in its experiments.

2.2. The paper identifies the following as one of the gaps in existing work: "Second, there is a lack of an explicit bias toward local changes." However, it is not justified how much this is needed for time-series classification.

2.3. "2.3 Multi-Resolution View Generation" => It is not clear which are adopted from the literature and which are new.

2.4. "the encoder focuses on learning time-domain local dynamics, while frequency information is computed directly from raw inputs and appended only at inference." => The paper does not elaborate how a network trained on time-domain data can work with time+frequency domain data without any retraining.

2.5. Not clear why the paper limits itself to univariate settings? The model can easily work with multivariate data.

3. Experimental evaluation needs to be improved.

3.1. Evaluation with just UCR Time Series Archive is limited.

3.2. "we use two complementary evaluation protocols." => If you follow prior work for this, please refer to the relevant literature. Otherwise, please justify/motivate why you need these protocols.

3.3. "To complement Protocol A, we conduct a focused comparison on eight representative datasets:" => How did you choose these eight datasets?

3.4. "train an RBF-kernel SVM on the extracted representations" => Why did you choose an SVM?

3.5. The considered baselines are relatively old. More recent approaches should be included, e.g.:

How Different from the Past? Spatio-Temporal Time Series Forecasting with Self-Supervised Deviation Learning, NeurIPS 2025.

3.6. The baselines should include prior Mamba-based contrastive learners.

4. The authors put forward efficiency as a primary feature of their method. However, this essentially stems from Mamba. I would suggest the authors to reevaluate their claims after including a comparison with other Mamba-based contrastive learners.

**Audience:**

Yes

**Audience Explanation:**

Self-supervised learning for time-series data can be of interest to the machine learning community.

**Broader Impact Concerns:**

The paper identifies efficiency as an important benefit of the paper, which is regarded positively by the reviewer.

**Claims And Evidence:**

No

**Claims Explanation:**

The paper has severe issues in its experimental evaluation. Please see Weaknesses under "Summary of Contributions".

**Requested Changes:**

Please see Weaknesses under "Summary of Contributions" as well as the following:

Minor comments:

- Fig 1 & 2: There are compression artifacts. I would suggest exporting the figure to PDF/SVG and using it as such.

- Eq 2: It is not clear whether h is the hidden representation of Mamba or its output.

- "Temperature schedule. For the temperature parameter \tau in InfoNCE, we apply cosine annealing ..." => Not sure whether this is the correct place for this. THe equation has not been introduced and this section talks about the Mamba encoder.

- Please number all your equations so that your readers can refer to them. For more details, see: https://wp.optics.arizona.edu/kupinski/wp-content/uploads/sites/91/2023/05/MerminEquations.pdf

- I strongly suggest the authors to use a different symbols for difference since the Delta symbol gives the impression of the paper is doing self-supervised learning involving the sampling rate (Delta) of Mamba.

- equation 5 => Equation 5. There are similar occurrences across the paper.

- "Let h denote either ~Delta or ~z" => You used h to denote the hidden states of Mamba (Eq 2). Please use a different symbol.

---

### Review · Reviewer_Yz28 · 2026-02-17

**Summary Of Contributions:**

This paper proposes DeltaSM, a self-supervised contrastive learning framework for univariate time-series representation that, as it were, combines three design elements in a manner the authors argue to be complementary: (i) a lightweight Mamba (selective state-space model) backbone that processes sequences in linear time, (ii) a delta-level contrastive objective operating on first-order differences of the latent sequence with curvature-adaptive (second-order) weighting to counterbalance Mamba's inherent smoothing tendency, and (iii) inference-time concatenation of frequency-domain descriptors (top-K FFT magnitudes) to the time-domain embeddings without incurring additional training cost. A token-budget scheme adaptively selects batch sizes to stabilize training across datasets of varying sequence length. The framework is evaluated on all 128 UCR archive datasets under two protocols: Protocol A (fixed 300-step budget, unified hyperparameters, SVM downstream) and Protocol B (8 selected datasets where baselines use their recommended settings while DeltaSM retains the Protocol A configuration). DeltaSM achieves the best mean accuracy (75.3%) and average rank (2.38) on UCR-128 under Protocol A while converging in a mean of 4.6 seconds, substantially faster than TS2Vec (23.2s), TS-TCC (14.0s), and TimesURL (79.6s). Ablations confirm the importance of curvature weighting and the delta-level loss.

**Key strengths:** The integrator-differentiator complementarity between the Mamba backbone and the delta-level objective is a well-motivated and clearly articulated design principle. The breadth of the evaluation (128 UCR datasets) under a strictly controlled compute budget is commendable and somewhat unusual in this subfield. The paper is generally well-organized with clear notation and helpful figures.

**Key weaknesses:** The evaluation is restricted entirely to univariate classification on UCR, with no multivariate, forecasting, or anomaly detection experiments. Under Protocol B, where baselines are given their recommended settings, DeltaSM (78.9%) falls behind TS2Vec (81.5%) and TimesURL (80.3%), somewhat tempering the headline claims. The frequency-feature integration at inference time, while effective, is a post-hoc concatenation that any baseline could equally adopt, making it difficult to attribute the gains specifically to the learned representations.

**Additional Comments:**

The paper is, on the whole, clearly written and well-structured, with a commendable level of transparency regarding experimental protocols and hyperparameter choices. The theoretical analysis in Section 4, while not providing formal guarantees, offers a cogent post-hoc interpretation of the empirical results that is consistent with recent analyses of SSM frequency bias (Wang et al., 2025; Yu et al., 2025). A few minor points:

- The name "DeltaSM" is somewhat ambiguous, as "Delta" could refer to the $\Delta$ parameter internal to Mamba's selective mechanism rather than the first-order differencing objective. A brief clarifying remark early in the paper would help.
- The UMAP visualization (Figure 5) is qualitatively informative but inherently subjective; quantitative cluster quality metrics (e.g., silhouette score) would add rigor.
- Table 7 in the appendix is an appreciated contribution to reproducibility. Providing the code and exact random seeds would further strengthen this aspect.

**Audience:**

Yes

**Audience Explanation:**

The intersection of state-space models and self-supervised learning for time series is a timely topic, and the question of how to reconcile computational efficiency with fine-grained temporal sensitivity is of broad interest to the time-series and representation learning communities. The observation that Mamba's smoothing tendency can be explicitly counteracted by a differentiation-based contrastive objective — framed as an "integrator-differentiator" complementarity — is a conceptually appealing insight that could inform future architecture-objective co-design beyond the specific framework proposed here. Furthermore, the comprehensive UCR-128 evaluation under a controlled compute budget fills a gap in the literature, as most prior Mamba-based time-series works have not been benchmarked at this scale. Practitioners working under resource constraints (e.g., edge deployment, rapid prototyping) would likely find the efficiency results directly useful.

**Broader Impact Concerns:**

The paper includes a Broader Impact Statement that is adequate in scope, acknowledging potential misuse in surveillance contexts and the need for careful validation in safety-critical domains. One additional concern that warrants brief mention is that the framework's emphasis on computational efficiency and rapid convergence could, if adopted without appropriate caution, encourage the deployment of time-series models in high-stakes settings (e.g., medical monitoring, predictive maintenance) with insufficient validation, given that the current evaluation is limited to classification benchmarks and does not assess calibration, uncertainty quantification, or robustness to distribution shift.

**Claims And Evidence:**

Yes

**Claims Explanation:**

The central claims of the paper — that DeltaSM achieves competitive or superior accuracy under a fixed, resource-constrained pretraining budget across the full UCR-128 archive, and that it does so with substantially reduced wall-clock time — are, on the whole, supported by the presented evidence with appropriate statistical testing (Friedman test with Nemenyi post-hoc). The critical difference diagram in Figure 2 provides convincing evidence that DeltaSM's average rank is meaningfully lower than competitors under Protocol A. The ablation studies in Tables 4 and 5 offer clear, controlled comparisons that isolate the contributions of curvature weighting, the delta-level loss, and the Mamba backbone, and the results are internally consistent with the paper's theoretical narrative.

That said, one must note several caveats. First, the Protocol A comparison is somewhat favorable to DeltaSM by design, as the 300-step budget is likely insufficient for the deeper or more complex baselines (e.g., TimesURL) to reach their intended operating regimes. The authors partially address this with Protocol B, but there DeltaSM does not achieve the best accuracy. Second, the frequency-feature concatenation is applied only to DeltaSM and not to baselines, introducing a confound; it would be more convincing if this post-hoc augmentation were also evaluated atop competitor representations. Third, the paper does not report confidence intervals or per-seed variance for the full 128-dataset evaluation, only for Protocol B, leaving some uncertainty about reproducibility. Despite these caveats, the overall evidence is sufficient to support the primary claims at the level of a workshop or journal contribution focused on efficiency.

**Requested Changes:**

The following changes are, in my estimation, necessary or beneficial for strengthening the submission:

1. **(Critical) Apply frequency-feature concatenation to baselines.** The current experimental design applies inference-time FFT features only to DeltaSM. Since this is a model-agnostic post-processing step, it should also be evaluated atop TS2Vec, TS-TCC, and TimesURL representations to determine how much of DeltaSM's accuracy advantage is attributable to the learned encoder versus the frequency augmentation. Without this control, the claimed accuracy gains are confounded.

2. **(Critical) Report per-seed variance on Protocol A.** The full UCR-128 results (Table 1, Table 7) report only mean accuracy across three seeds without standard deviations or confidence intervals. Given the large number of datasets and modest differences in mean accuracy between methods (e.g., DeltaSM 0.753 vs. 1-NN DTW 0.728), per-seed variance is necessary to assess statistical reliability.

3. **(Strongly recommended) Evaluate on at least one non-classification task.** The paper repeatedly claims that DeltaSM learns "general-purpose" and "transferable" representations, yet evaluation is limited to classification on UCR. Including even a preliminary evaluation on anomaly detection (e.g., using the UCR anomaly archive or Yahoo/Numenta benchmarks) or forecasting would meaningfully support the generality claim and is needed to justify the framing.

4. **(Strongly recommended) Include multivariate datasets.** The restriction to univariate series is a notable limitation for a framework that aspires to generality. Even a small-scale evaluation on a subset of the UEA multivariate archive would help assess whether the delta-level objective and Mamba backbone retain their advantages in the multivariate setting.

5. **(Recommended) Clarify Protocol A fairness more carefully.** The 300-step budget is motivated as a controlled comparison, but different architectures converge at different rates by design. A complementary evaluation where all methods are given sufficient steps to converge (even if DeltaSM converges first) and then accuracy is compared at convergence would help disentangle "learns faster" from "learns better."

6. **(Minor) Discuss the sensitivity to $K$ (number of FFT bins).** The choice $K = 64$ is fixed without justification or ablation. Since this feature is concatenated at inference and contributes meaningfully to accuracy, its sensitivity deserves brief treatment.

7. **(Minor) The token-budget heuristic (Section 2.2) involves several magic numbers** ($B_{\text{tok}} = 32768$, factor of 1.5, percentile90, etc.) that are not ablated. A brief sensitivity analysis or at minimum a discussion of how these values were chosen would be helpful.

---

### Review · Reviewer_bLef · 2026-03-22

**Summary Of Contributions:**

The paper presents a technically sound engineering effort, integrating a Mamba backbone with a tailored contrastive objective. Its core research contribution is undermined by a significant misalignment with the current trajectory of the field.

**Audience:**

No

**Audience Explanation:**

The research question is largely outdated.
The paper frames its motivation around the challenge of learning transferable representations from scratch on individual, label-scarce datasets (e.g., training on one UCR dataset at a time). However, the field of time-series analysis has moved decisively toward large-scale pre-trained foundation models (e.g., Timer, UniTime, TimesFM, MOIRAI). These models leverage large, diverse corpora to learn highly generalizable representations via zero-shot or few-shot transfer, rendering the approach of training separate contrastive models per dataset (even if efficient) less relevant. The premise that “Mamba is underexplored in this setting” does not constitute a sufficient contribution when the setting itself is being superseded by more scalable paradigms.

**Claims And Evidence:**

No

**Claims Explanation:**

The validation scope is far too narrow to support the claim of “general-purpose” representation learning.
The authors evaluate their representations exclusively on classification tasks (UCR archive). Self-supervised representation learning, as established by prior work cited in the abstract (TS2Vec, TS-TCC), is expected to serve a variety of downstream tasks, including forecasting, anomaly detection, and imputation. Evaluating solely on classification, even across 128 datasets, tests only one dimension of representation quality (linear separability). A framework claiming to reconcile “efficiency and expressivity” for general-purpose representations must demonstrate utility beyond classification. The absence of forecasting and other tasks leaves the robustness and versatility of the learned representations unsubstantiated.

**Requested Changes:**

There are two main concerns:

1. The validation scope is far too narrow to support the claim of “general-purpose” representation learning.

The authors evaluate their representations exclusively on classification tasks (UCR archive). Self-supervised representation learning, as established by prior work cited in the abstract (TS2Vec, TS-TCC), is expected to serve a variety of downstream tasks, including forecasting, anomaly detection, and imputation. Evaluating solely on classification, even across 128 datasets, tests only one dimension of representation quality (linear separability). A framework claiming to reconcile “efficiency and expressivity” for general-purpose representations must demonstrate utility beyond classification. The absence of forecasting and other tasks leaves the robustness and versatility of the learned representations unsubstantiated.

2. The research question is largely outdated.

The paper frames its motivation around the challenge of learning transferable representations from scratch on individual, label-scarce datasets (e.g., training on one UCR dataset at a time). However, the field of time-series analysis has moved decisively toward large-scale pre-trained foundation models (e.g., Timer, UniTime, TimesFM, MOIRAI). These models leverage large, diverse corpora to learn highly generalizable representations via zero-shot or few-shot transfer, rendering the approach of training separate contrastive models per dataset (even if efficient) less relevant. The premise that “Mamba is underexplored in this setting” does not constitute a sufficient contribution when the setting itself is being superseded by more scalable paradigms.

---

### Note · Authors · 2026-03-23

**Comment:**

Dear Reviewers and Editors,

Thank you very much for your careful and thoughtful reviews. After carefully considering the feedback, I have decided to withdraw the manuscript.

In particular, I take seriously the concerns that the current empirical validation is too limited to support the broader claims of general-purpose representation learning, and that the paper needs a clearer articulation of its central contribution and positioning. Addressing these issues properly would require more than incremental revisions or additional experiments; it would require a substantial reconsideration of the problem framing, claims, and evaluation design. I do not believe that such a revision can be completed responsibly within the current revision period.

For this reason, I believe it is more appropriate to withdraw the paper at this stage and revisit the work only after a more fundamental redesign of the method, its positioning, and its validation.

I sincerely appreciate the reviewers’ time and constructive feedback.

**Withdrawal Confirmation:**

I have read and agree with the venue's withdrawal policy on behalf of myself and my co-authors.